# Full-length transcript characterization of *SF3B1* mutation in chronic lymphocytic leukemia reveals downregulation of retained introns

Alison D. Tang 🄳 [1], Cameron M. Soulette 🄳 [2], Marijke J. van Baren[1], Kevyn Hart[1], Eva Hrabeta-Robinson[1], Catherine J. Wu 🄳 [3,4,5] & Angela N. Brooks 🄳 [1✉]

While splicing changes caused by somatic mutations in *SF3B1* are known, identifying full-length isoform changes may better elucidate the functional consequences of these mutations. We report nanopore sequencing of full-length cDNA from CLL samples with and without *SF3B1* mutation, as well as normal B cell samples, giving a total of 149 million pass reads. We present FLAIR (Full-Length Alternative Isoform analysis of RNA), a computational workflow to identify high-confidence transcripts, perform differential splicing event analysis, and differential isoform analysis. Using nanopore reads, we demonstrate differential 3' splice site changes associated with *SF3B1* mutation, agreeing with previous studies. We also observe a strong downregulation of intron retention events associated with *SF3B1* mutation. Full-length transcript analysis links multiple alternative splicing events together and allows for better estimates of the abundance of productive versus unproductive isoforms. Our work demonstrates the potential utility of nanopore sequencing for cancer and splicing research.

[1] Department of Biomolecular Engineering, University of California, Santa Cruz, CA 95062, USA. [2] Department of Molecular Cell & Developmental Biology, University of California, Santa Cruz, CA 95062, USA. [3] Department of Medical Oncology, Dana-Farber Cancer Institute, Boston, MA 02215, USA. [4] Broad Institiute of Harvard and MIT, Cambridge, MA, USA. [5] Department of Medicine, Brigham and Women's Hospital, Harvard Medical School, Boston, MA 02115, USA. ✉email: anbrooks@ucsc.edu

In various cancers, mutations in the splicing factor *SF3B1* have been associated with characteristic alterations in splicing. In particular, recurrent somatic mutations in *SF3B1* have been linked to various diseases, including chronic lymphocytic leukemia (CLL)[1–4], uveal melanoma[5–7], breast cancer[8–10], and myelodysplastic syndromes[11,12]. *SF3B1* is a core component of the U2 snRNP of the spliceosome and associates with the U2 snRNA and branch point adenosine of the pre-mRNA[13–15]. In CLL, mutations in the HEAT-repeat domain of *SF3B1*, such as the K700E hotspot mutation, have been shown to associate with poor clinical outcome[1–4]. B-cell-restricted expression of *SF3B1* mutation together with *Atm* deletion leads to CLL-like disease at low penetrance in a mouse model, confirming a contributory driving role of mutated *SF3B1*[16]. In addition, mutations in *SF3B1* induce aberrant splicing patterns that have been well-characterized using short-read sequencing of the transcriptome. Most notably, mutant *SF3B1* has been shown to generate altered 3′ splicing[17–19]. Mutant *SF3B1*-associated changes in branch point recognition and usage[20–22] form the model in which mutant *SF3B1* affects acceptor splice sites. Targeting a mutant *SF3B1*-induced mis-spliced exon in the tumor suppressor *BRD9* using antisense oligonucleotides suppresses tumor growth[23], further revealing the therapeutic implications for treating mutant *SF3B1*.

While alternative splicing (AS) patterns induced by *SF3B1* mutations have been examined on an event level with short reads, these patterns have not been systematically examined on an isoform level. Studying splicing factor mutations at the level of single splice junctions limits our complete understanding of the functional consequences of these aberrant splicing changes. Short reads provide an incomplete picture of the transcriptome because exon connectivity is difficult to determine from fragments[24]. Moreover, detection and quantification of transcripts containing retained introns using short reads is difficult and often overlooked[25]. Long-read-sequencing techniques now offer increased information on exon connectivity by sequencing full-length transcript molecules[26–28]. Nanopore sequencing works by measuring the change in electrical current caused by DNA or RNA threading through a nanopore and converting the signal into nucleotide sequences[29,30]. Nanopore sequencing yields reads as long as 2 megabases[31] and has been used for applications ranging from the sequencing of the centromere of the Y chromosome[32], the human genome[33], and single-cell transcriptome sequencing[34]. However, nanopore technology has yet to be thoroughly explored as a tool for differential splicing, nor has it been used to explore cancer-associated splicing factor mutations. The high-throughput Oxford Nanopore PromethION has the potential to yield upward of 100 gigabases (Gb) per flow cell[35]; the substantial number of long molecules sequenced makes the PromethION applicable to our purposes of transcript detection and quantification.

To investigate *SF3B1^{K700E}* AS at an isoform level with nanopore data, the representative transcripts and their abundance in each condition must be determined from the reads. Existing software for short-read RNA-Seq data that perform isoform assembly, splicing, or quantification analyses are not designed to work properly with the length of and frequent indels present in nanopore reads. The raw accuracy of nanopore 1D cDNA sequencing is ~85–87%[36–38], although accuracy can change depending on iterations of the technology and library preparation methods[37]. In contrast, the long consensus reads produced by Pacific Biosciences are able to achieve much higher base accuracies[39]. Thus, software tools developed for PacBio sequencing were not developed for noisier data and may disregard raw reads with less than 99% accuracy[40] or discard assembled isoforms with too many errors[41]. To assemble isoforms and perform splicing analysis from nanopore reads, we have created a workflow called Full-Length Alternative Isoform analysis of RNA (FLAIR). FLAIR requires a reference genome to define isoforms from long reads. While FLAIR does not require short reads, having matched short-read data can be used to identify unannotated splice sites and improve the confidence of transcript splice junction boundaries. Recognizing the benefit of highly accurate short reads for detecting the splice junctions of a mutated splicing factor, we used a hybrid-seq approach in this study. We combined the accuracy of Illumina short reads for splice junction accuracy with the exon connectivity information of long reads to overcome the higher error rates of long reads.

Of a large cohort of CLL patient tumor samples characterized using short-read RNA-Seq[17], we present the resequencing of a subset of these transcriptomes, globally, with nanopore technology: three with wild-type *SF3B1*, three with the K700E mutation, and three additional normal B-cell samples, which are the normal lineage cellular complement to CLL, to use as a normal tissue control[42]. Following the identification of high-confidence isoforms from nanopore data, FLAIR provides a framework for performing AS and differential isoform usage analyses. Upon splicing analysis of the nanopore CLL data, we observe a bias toward increased alternative 3′ splice sites over alternative 5′ splice sites in CLL *SF3B1^{K700E}* samples, consistent with the known effects of *SF3B1* mutation. We also highlight a previously underappreciated finding of differential intron retention in CLL *SF3B1^{K700E}* versus *SF3B1^{WT}* with increased splicing relative to wild-type *SF3B1* samples. Using long reads, we can identify retained introns more confidently than with short reads and are able to observe AS events across full-length isoforms. FLAIR analysis of nanopore data reveals biological insights into *SF3B1* mutations and demonstrates the potential for discovering cancer biology with long-read sequencing.

## Results

**FLAIR identifies full-length isoforms in CLL.** As a pilot study to characterize *SF3B1^{K700E}* full-length transcripts, we performed nanopore 2D cDNA minION sequencing of a CLL *SF3B1^{K700E}* sample that was clonal for the mutation and a CLL *SF3B1^{WT}* sample that did not contain other common CLL-associated mutations. These samples were previously examined in a study of AS of *SF3B1* mutants using short-read RNA-Seq data[17]. With the pilot nanopore data, we found an increase in aberrant 3′ splicing in CLL *SF3B1^{K700E}* (Supplementary Fig. 1), recapitulating findings from the CLL short-read data[17]. This motivated the resequencing of additional replicate samples with nanopore 1D cDNA sequencing on the PromethION, a high-throughput nanopore sequencer (Fig. 1a). From the same cohort, we sequenced six primary CLL samples and three B cells, generating 257 million total reads with large variability in read depth and percentage of pass reads for each flow cell (Table 1). On average, 30.5% of the PromethION reads were considered full-length ("Methods"). Although the RNA samples had been stored for 5 years, they showed minimal signs of degradation as the read lengths were comparable to those observed from other nanopore cDNA-sequencing runs[26,38,43]. We observed relatively strong gene expression correlation (average Pearson correlation 0.822) between the three samples sequenced on both the minION and promethION (Supplementary Fig. 2), and so combined the data for subsequent analyses.

We developed FLAIR to generate a set of high-confidence isoforms that were expressed in our samples. FLAIR summarizes nanopore reads into isoforms in three main steps: alignment, correction, and collapse (Fig. 1b). In the alignment step, we aligned raw read sequences from all samples to the genome to identify the general transcript structure. We compared the long-read spliced-aligners minimap2[44] and GMAP[45], the latter of

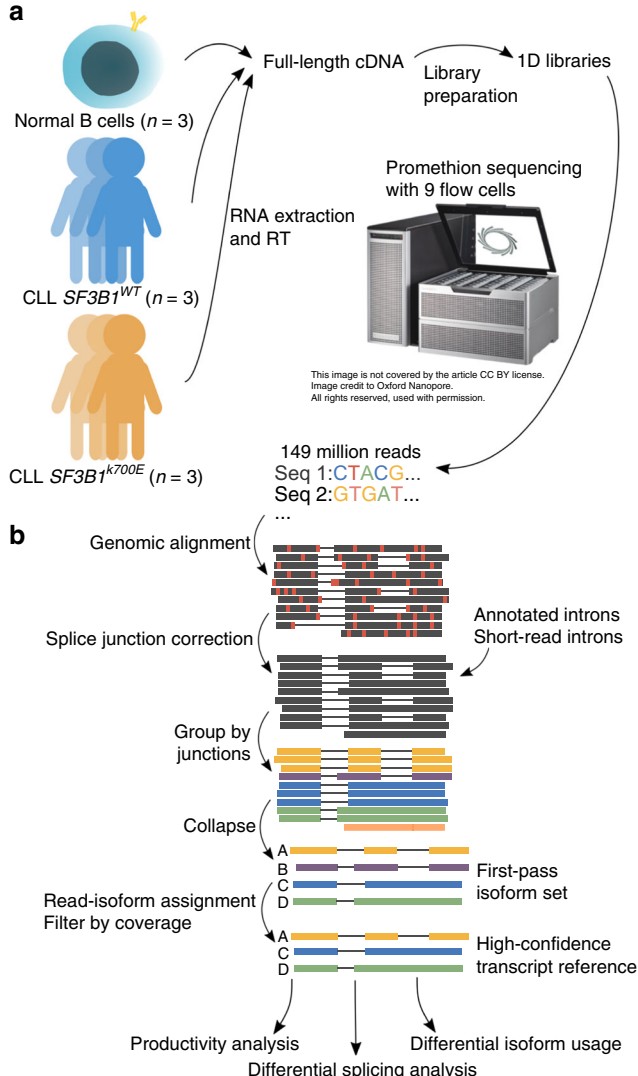

**Fig. 1 Long-read nanopore sequencing and FLAIR analysis to identify full-length transcripts associated with *SF3B1* mutation in chronic lymphocytic leukemia. a** RNA from primary samples across three conditions (chronic lymphocytic leukemia with and without *SF3B1* mutation and normal B cells) were obtained. The RNA was prepared into 1D cDNA libraries and each sample was sequenced on a PromethION flow cell. The basecalled data were processed using the FLAIR pipeline. **b** The FLAIR pipeline constructs an isoform set from nanopore reads. First, reads are aligned to the genome with a spliced aligner. The sequence errors are marked in red. Next, they are splice-corrected using splice sites from either annotated introns, introns from short-read data, or both. The corrected reads are grouped by their splice junction chains and are summarized into representative isoforms (first-pass set). All reads are then reassigned to a first-pass isoform. The isoforms that surpass a supporting read threshold of 3 comprise the final high-confidence isoform set.

which has been used in several other long-read studies[34,46,47]. The aligners were evaluated on a subset of the CLL data according to splice-site accuracy, comparing splice sites mapped by each aligner with annotated splice sites. Minimap2 demonstrated marked improvement in splice-site mapping over GMAP (Fig. 2a).

The next main step of FLAIR is to *correct* splice sites because the relatively high-sequencing error rates, frequent base deletions, and difficulties of spliced alignment can result in spurious alignments near splice sites (Supplementary Fig. 3). To address

indels, small gaps in the read alignments were artificially filled ("Methods"). FLAIR only considers correct splice sites as those from a curated set of splice junctions, such as splice sites present in annotations. To correct splice sites, an incorrect splice site in a read alignment is replaced with a correct one as long as the correct splice site is within a window size of 10 bp. Given that *SF3B1* mutations can subtly alter 3′ splice-site choice[17–19,22] and accurate representation of splice-site usage is crucial, we increased splice-site accuracy by correcting read alignments using splice sites in either annotations or matched CLL short-read sequencing (Fig. 2a, Supplementary Fig. 3c, d). We initially identified many minimap2-aligned splice sites that aligned outside of the window size from the nearest supported splice site, were unannotated, and had no short-read support; we determined that these were of lower confidence based upon further manual inspection. Many of these novel sites appeared to be driven by alignment errors. Thus, we did not consider nanopore reads with novel splice sites that had no additional support.

The *collapse* step of FLAIR produces a set of expressed isoforms with high confidence for further downstream analysis. Starting with only the fully splice-corrected reads, FLAIR constructs a first-pass nanopore isoform transcriptome, collecting the reads with the same unique splice junctions chain into distinct isoform groups (Fig. 1b). FLAIR determines one or more representative isoform(s) within each group by calling confident transcription start and end sites based on the density of read start and end positions ("Methods"). Next, FLAIR reassigns all of the reads to a first-pass isoform, including reads with minimap2-aligned splice sites that were not able to be fully splice corrected. This is done by aligning the reads to spliced sequences from the first-pass isoform set (i.e., aligning reads to an isoform sequence without a spliced alignment) and assigning the read to an isoform with the best alignment with MAPQ ≥ 1. This realignment of reads to the set of nanopore-specific isoforms accounted for misalignments that manifested from spliced alignment by constraining reads to align only to splice junction chains with additional support. The realignment was also crucial for better distinguishing splice-site differences[17]. Finally, FLAIR filters out first-pass isoforms that have fewer than three supporting reads and the remaining isoforms with sufficient coverage comprises the final high-confidence nanopore isoform set. Using FLAIR, we identified a total of 326,699 high-confidence spliced isoforms (Supplementary Data 1, 2). Of these isoforms, 32,479 matched annotated isoforms and the majority (90.0%) were unannotated. Most of the unannotated isoforms were a novel combination of already annotated splice junctions (142,971), while others deviated from the annotation because they contained a retained intron (21,700) or a novel exon (3594). The remainder of the unannotated isoforms contain at least one novel splice site not present in annotations but supported through short reads. We performed a saturation analysis of the reads, by condition, to assess the number of FLAIR isoforms that could be detected at differing read depths (Supplementary Fig. 4). For all conditions, we found that sequencing at greater depth would facilitate the discovery of even more isoforms. At all read depths, we were able to detect the greatest number of isoforms in CLL $SF3B1^{K700E}$. The read lengths of CLL $SF3B1^{WT}$ were noticeably shorter, resulting in fewer isoforms able to be detected compared with B cell or CLL $SF3B1^{K700E}$. The saturation analysis illustrates the diversity of isoforms resulting from mutant *SF3B1* as well as the importance of read length for isoform discovery.

**FLAIR improves isoform detection from nanopore reads.** We evaluated FLAIR against other nanopore isoform assembly methods: Mandalorion[34] and Oxford Nanopore's Pinfish

**Table 1 Nanopore-sequencing statistics.**

| Sample | Data (Gb) | Total reads | % Pass | % full length | Median pass read length | Mean pass read length | No. of genes detected | % genes with AS |
|---|---|---|---|---|---|---|---|---|
| Promethion WT 1 | 2.75 | 5,839,921 | 23.1 | 22.9 | 816 | 1020 | 17,369 | 53.0% |
| Promethion WT 2 | 0.526 | 2,051,438 | 6.91 | 27.7 | 768 | 936 | 8492 | 43.9% |
| Promethion WT 3 | 52.2 | 63,378,471 | 56.9 | 28.0 | 788 | 944 | 43,223 | 41.7% |
| Promethion MT 1 | 0.522 | 3,016,571 | 1.70 | 24.0 | 718 | 862 | 6272 | 31.8% |
| Promethion MT 2 | 91.1 | 93,130,856 | 74.8 | 30.2 | 870 | 1020 | 45,744 | 41.2% |
| Promethion MT 3 | 3.1 | 15,101,288 | 5.38 | 30.2 | 919 | 1123 | 16,218 | 54.7% |
| Promethion B cell 1 | 65.8 | 66,311,147 | 60.7 | 34.3 | 948 | 1099 | 33,538 | 46.6% |
| Promethion B cell 2 | 0.970 | 5,390,853 | 7.16 | 33.5 | 712 | 951.3 | 13,638 | 44.0% |
| Promethion B cell 3 | 0.731 | 2,934,130 | 4.80 | 27.8 | 766 | 1015 | 9372 | 34.6% |
| Minion WT 1 | 0.105 | 119,091 | – | 19.6 | 651 | 868 | 8320 | 36.6% |
| Minion MT 1 | 0.138 | 129,798 | – | 25.5 | 880 | 1084 | 9747 | 42.6% |
| Minion B cell 1 | 0.042 | 46,077 | – | 32.5 | 749 | 922 | 5379 | 26.8% |

For each run, the following statistics were computed: gigabase of data, total reads, the percentage of reads designated as passing by the basecaller for PromethION runs, percentage of aligning pass reads that are full-length, median pass read length, mean pass read length, the number of genes covered, and the number of genes with evidence of alternative splicing ("Methods"). For the 2D MinION runs, the total number of reads is the number of 2D reads, which was also considered the total number of pass reads.

(https://github.com/nanoporetech/pinfish). We compared different thresholds for the minimum number of supporting reads per isoform to determine the number of reads necessary to build isoforms accurately. We benchmarked using GFFCompare[48] on assemblies using both a simulated dataset and Lexogen Spike-In RNA Variants (SIRV)[37] sequenced with nanopore 1D technology. We simulated a set of nanopore transcriptome reads by creating a wrapper for NanoSim[49], a tool created for simulating genomic nanopore reads ("Methods"). The SIRV RNAs are of known sequence and concentration and are of comparable complexity to human transcripts[50]. We did not benchmark FLAIR with short-read data as Mandalorion and Pinfish do not incorporate short reads into their algorithms for a splice-site correction step. Instead, Mandalorion uses racon[51] to generate consensus isoform sequences from clustered reads. Mandalorion was initially designed for 2D reads[34], however, has been reworked to perform best on higher-accuracy R2C2 reads[37]. Pinfish clusters nanopore reads and generates consensus isoforms using the median exon boundaries from all transcripts in the cluster. We observed that FLAIR had higher sensitivity than other methods and had comparable or higher precision (Fig. 2b and Supplementary Fig. 5). Based on the base-level and intron chain-level sensitivities, Pinfish appeared to require more reads in order to determine isoform structures. We also noted that as the supporting read threshold was increased, there was an expected decrease in sensitivity and negligible increase in precision. As a result, we decided to use FLAIR with a threshold of 3, increasing the number of isoforms assembled to better capture all splicing alterations.

Quantifying expression of isoforms using nanopore data may appear straightforward in that each nanopore read corresponds to a single transcript present in the cell. However, ambiguity of read assignments to transcripts can occur when assigning noisy, truncated reads to genes with greater isoform complexity. We benchmarked different approaches of assigning reads to transcripts using the 1D SIRV reads and the simulated nanopore reads using either a transcriptome-alignment approach or salmon[52]. The former involved aligning reads to transcript sequences using minimap2 and selecting alignments using either primary read alignments or read alignments with mapping quality scores greater than or equal to 1. The salmon algorithm is able to account for sequence-specific, fragment-GC, and positional biases. We evaluated salmon as a quantification tool using its alignment-based mode with all minimap2 alignments, as

performance of the quasi-alignment mode could suffer from low read accuracy. We observed that counting transcript alignments with MAPQ ≥ 1 correlated the best with expected counts for both the SIRV and simulated data (Fig. 2c) and used this quantification approach for our analyses. Although the correlations using salmon were not as strong as the MAPQ ≥ 1 approach, the latter approach filters out many low-quality reads and is unable to include them in the counts; in the SIRVs, 25.5% (5,469 out of 21,456) of reads have MAPQ of 0 and were discarded from quantification. We recognized the increased inclusion of reads from the data when using salmon and provided an option to use salmon with FLAIR.

**Nanopore read analysis detects 3′ splice-site changes.** Previous studies have demonstrated that $SF3B1^{K700E}$ promotes alternative 3′ splice-site (3′SS) usage[17], a pattern we sought to validate in our nanopore data. From a cohort of 37 CLL samples[17] (24 $SF3B1^{WT}$ and 13 with $SF3B1$ mutation) sequenced with Illumina short reads, 65 significantly altered 3′ splicing events associated with $SF3B1^{K700E}$ were identified using juncBASE[53]. For those significant events, we measured the change in percent spliced-in (dPSI) values using the corrected nanopore reads by subtracting the PSI of CLL $SF3B1^{WT}$ from the PSI of CLL $SF3B1^{K700E}$ and compared them with the short-read dPSIs (Fig. 3a). The long-read dPSIs were correlated with the CLL short-read dPSIs, and dPSIs were more similar across the two sequencing technologies with increasing long-read depth (Pearson correlation 0.952). Some splice junctions had insufficient coverage in our nanopore data for adequate power to detect the same splice-site usage observed with short reads; we therefore wanted to identify the altered 3′ splicing events that would be significantly altered by mutant $SF3B1$ in the nanopore data.

Due to the sheer complexity of alternatively spliced isoforms, we wrote an AS event caller for FLAIR to more accurately classify AS in isoforms ("Methods"). We called alternative 3′ and 5′ splice sites observed in the FLAIR isoforms and quantified the coverage of the alternative events in each of the CLL samples. Using DRIMSeq[54] for statistical testing, we identified 35 alternative 3′ SSs and 10 alternative 5′SSs that were significantly differentially spliced (corrected $p$ value < 0.1 and dPSI absolute value > 10) between $SF3B1^{K700E}$ and $SF3B1^{WT}$ (Supplementary Data 3). More $SF3B1^{K700E}$-associated 3′ alterations were upstream of $SF3B1^{WT}$-associated 3′SSs (20 out of 35) and only 2 of the 35 alternative 3′ SSs had been previously identified with short-read sequencing.

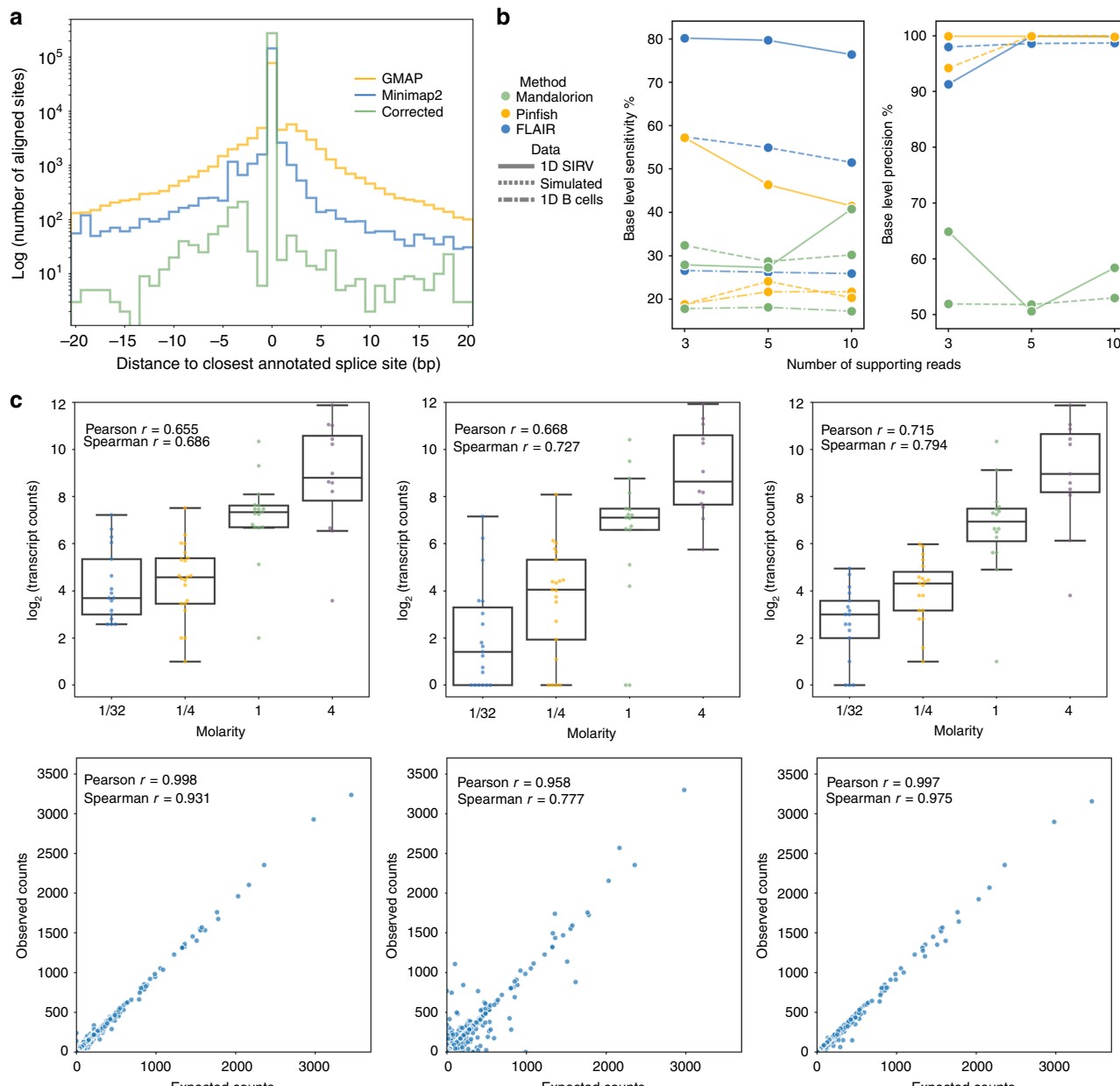

**Fig. 2 Evaluation of nanopore read alignment, correction, transcript assembly, and transcript quantification. a** Histogram showing the distance between splice sites determined using the GMAP (yellow) or minimap2 (blue and green) spliced aligners to the closest annotated splice sites for a subset of the CLL SF3B1$^{K700E}$ reads; the minimap2 aligned vs annotated splice-site histogram was further distinguished between alignments that had (green) or had not (blue) been subject to FLAIR splice correction. **b** Base-level sensitivity (left) and precision (right) of isoform sets built using Mandalorion (green), Pinfish (yellow), or FLAIR (blue). Each software tool was run with varying supporting read thresholds (*x*-axis) on either nanopore-sequenced 1D SIRVs (solid line), a set of simulated nanopore transcript reads (dashed), or 1D B cell reads subsampled down to 500,000 reads (dot dash). **c** Comparison of transcript quantification methods for nanopore reads plotting the expected and observed transcript abundances of the SIRV data (top row) and simulated data (bottom row). First column: counting primary alignments, second column: salmon given all alignments, third column: counting MAPQ ≥ 1 alignments. Box-plots show median line, box limits are upper and lower quartile, and whiskers are 1.5× interquartile.

We were unable to find any unifying sequence motif associated with these altered 3′SS identified in the nanopore data. Using the 65 alternative 3′SSs significantly associated with *SF3B1* mutation identified in the CLL short-read data, we found a tract of As 13–16 bp upstream of the canonical 3′SS (Supplementary Fig. 6a). This motif is concordant with other mutant *SF3B1* studies using short reads[18],[19]. From the nanopore-sequencing analysis, the distribution of distances between *SF3B1*$^{K700E}$-altered 3′SSs to canonical sites peaks around −20 bp and is significantly different from a control distribution (two-sided Mann–Whitney *U* $p = 6.77 \times 10^{-2}$)

(Fig. 3b), similar to what has been reported in CLL short-read sequencing[17]. One of the alternative 3′SS identified from both long and short reads was in the *ERGIC3* gene (Fig. 3c). There were two dominant isoforms: a novel isoform containing the proximal splice site that was more abundant in *SF3B1*$^{K700E}$ and another annotated isoform containing the distal splice that was expressed in both the mutant and wild type samples. Both the proximal and distal 3′SS were associated with multiple isoforms with distinct AS patterns up- and downstream of the alternative 3′SS. Long reads enabled us to not only identify mutant *SF3B1*-altered splice sites, but also

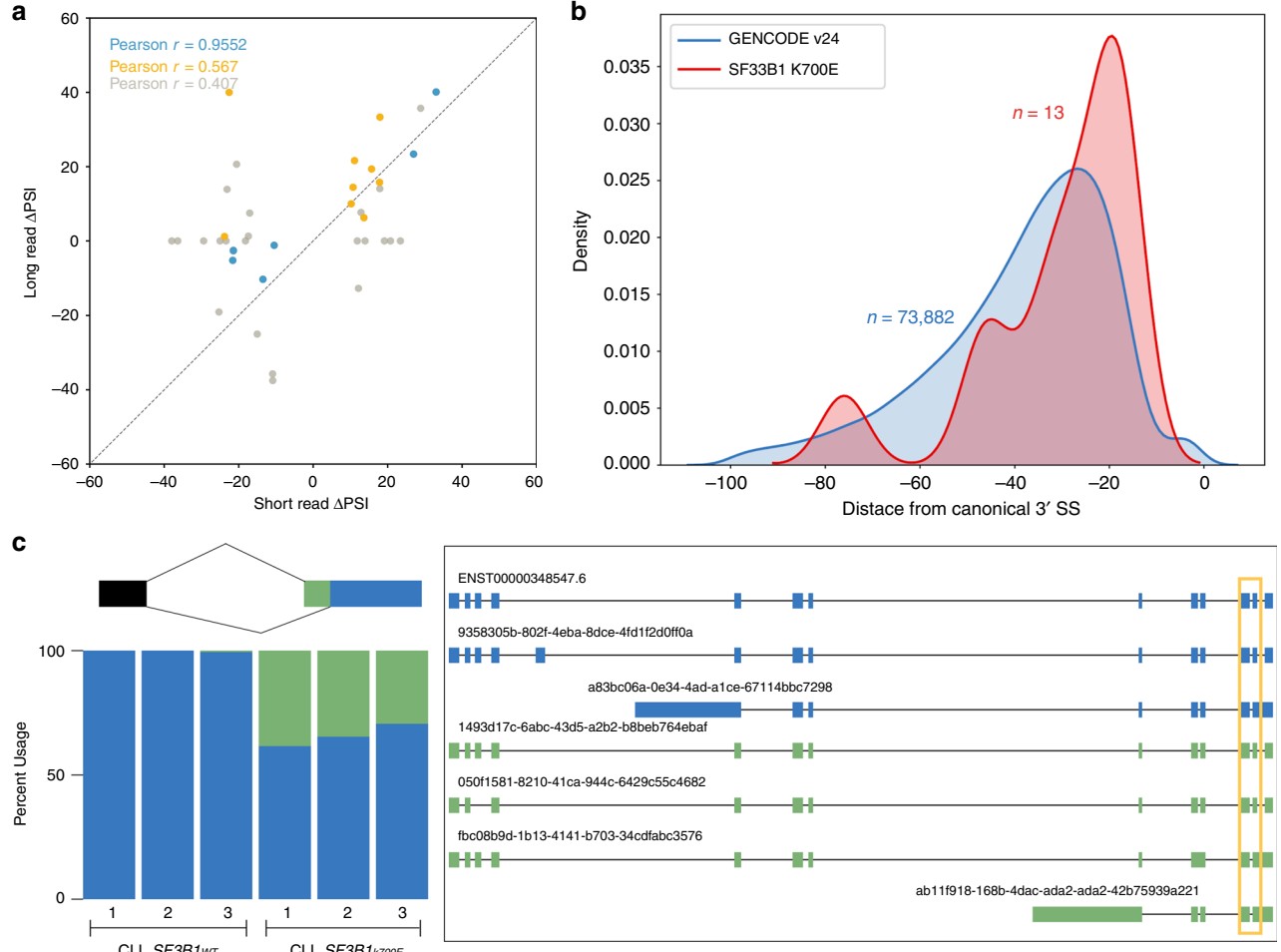

**Fig. 3 Alternative 3′ splicing patterns identified with nanopore-sequencing data are concordant with short read data and reveal additional splicing complexity. a** Comparison of the delta PSI for significant alternative splicing events identified by juncBASE using short-read data and the median delta PSI of the same events using long-read data. The colors correspond to the median coverage of the splice junctions, where blue is greater than 25 reads, yellow is greater than 10, and gray is greater than 0. **b** Distribution of proximal splice sites found in nanopore reads for 13 significant sites (corrected *p* value < 0.10, delta PSI > 10). The GENCODE distribution is the distribution of distances from canonical 3′ splice sites to the nearest non-GAG trimer. Source data are provided in a Source Data file. **c** *ERGIC3* splice-site usage (left) and full isoforms (right) for the proximal chr20:35,556,954 (green) and distal chr20:35,556,972 (blue) sites from 5′ to 3′. The dominant isoforms using either the proximal or distal site are 1493d17c-6abc and ENST00000348547.6. The alternative acceptor event is boxed in yellow in the isoform schematic.

associate an event-level aberration with full-length isoforms containing other alternative processing events.

**SF3B1^K700E downregulates intron retention events**. Intron retentions (IR) have been observed to differentiate tumors from matched normal tissue, as they are highly prevalent across a variety of cancers[55,56]. However, based on common approaches used by short-read AS analysis tools, it is difficult to characterize IR event usage confidently using short reads[25]. Thus, unless stringent approaches are used, intron retention events are easily misclassified particularly in regions with complex AS[57]. With long reads, a single read is capable of connecting multiple AS events in addition to spanning IR, enabling easier identification and quantification of IR. To investigate changes in IR associated with *SF3B1^K700E*, we categorized each FLAIR isoform as IR-containing or not (spliced). Comparing the expression fold-change between CLL samples revealed that IR isoforms were globally downregulated in the *SF3B1^K700E* sample compared with CLL *SF3B1^WT* (Fig. 4a). When performing the same comparison between B cell and *SF3B1^K700E*, we observed no significant difference in the expression of IR-containing isoforms (two-sided

Mann–Whitney *U p* = 0.121). Reanalysis of the CLL short-read data confirmed the observed increase in the inclusion of retained introns in CLL *SF3B1^WT* samples (Fig. 4b).

To further investigate the effect of *SF3B1^K700E* on increased intron splicing, we reanalyzed Nalm-6 Pre-B isogenic cell lines[19] with either *SF3B1^WT* or *SF3B1^K700E* sequenced using short reads. We used juncBASE[53] to identify and quantify AS between the two conditions. For the 16 significant (corrected *p* < 0.1) IR events, Nalm-6 *SF3B1^K700E* PSI values appeared lower than *SF3B1^WT* (Supplementary Fig. 7a), supporting a decrease in retained introns in *SF3B1^K700E*-containing samples; however, the difference was not significant. We observed that for IRs that were more spliced with mutant *SF3B1*, they were spliced with greater magnitude than the IRs more spliced in the wild type (Supplementary Fig. 7b). In addition, we reanalyzed TCGA breast cancer samples without common splicing factor mutations against samples with *SF3B1^K700E* using juncBASE and found the same trend of increased IR splicing in *SF3B1^K700E* (Supplementary Fig. 7c).

Seeing that the trend of higher IR expression in *SF3B1^WT* was observed transcriptome-wide in the nanopore data, we narrowed

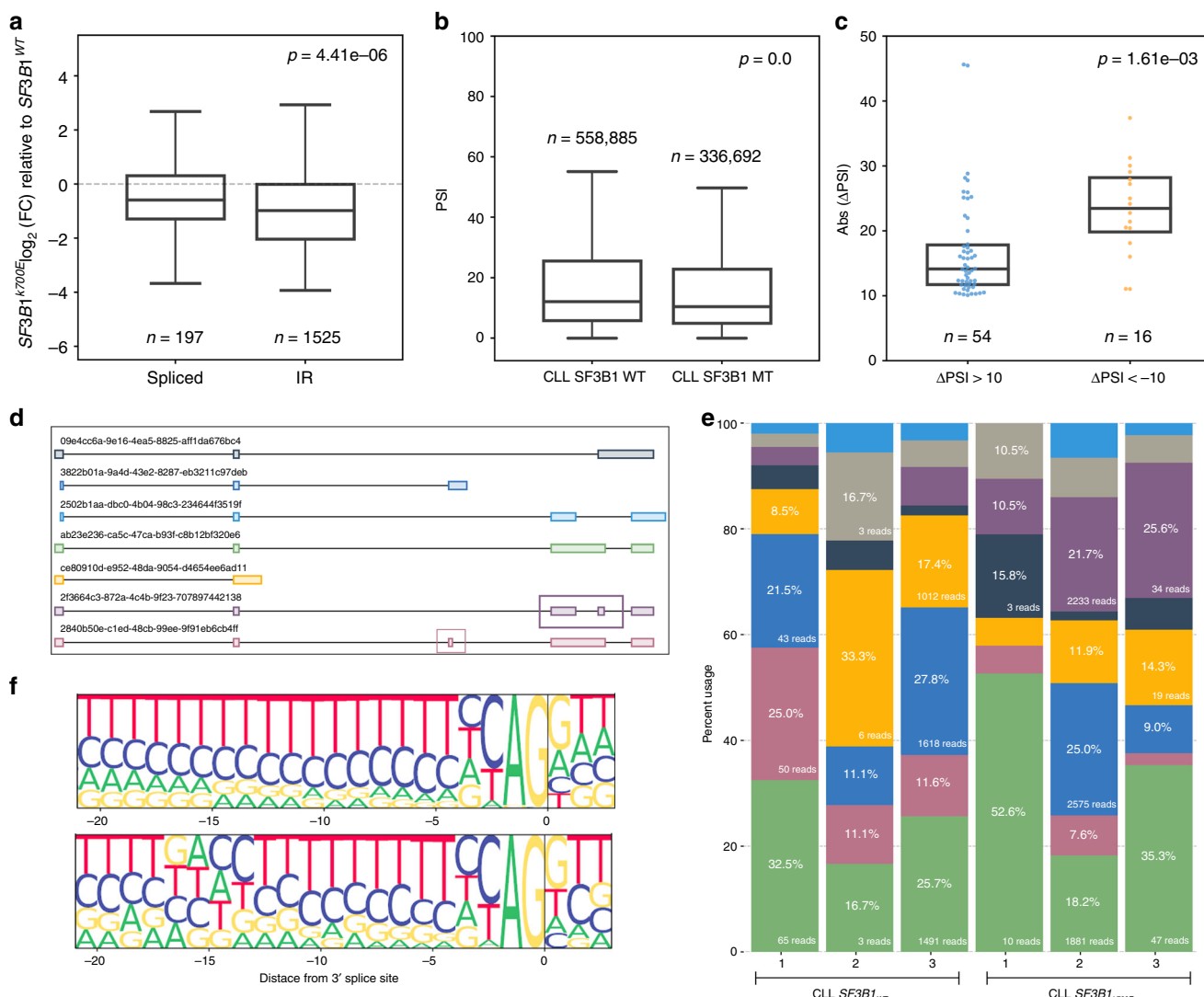

**Fig. 4 Intron retention events are more strongly downregulated in CLL *SF3B1*$^{K700E}$. a** Expression fold-change between SF3B1$^{K700E}$ and SF3B1$^{WT}$ of FLAIR isoforms with (IR) or without (spliced) retained introns. Boxplot median difference = 0.395. Source data are provided in a Source Data file. **b** PSI values of intron retention events identified in short read sequencing of CLL *SF3B1* WT or CLL *SF3B1* MT samples. Boxplot median difference = 1.69. Source data are provided in a Source Data file. **c** The change in PSI in significant intron retention events (corrected $p < 0.1$) identified in nanopore data that are more included in CLL *SF3B1*$^{K700E}$ (blue) or more included in CLL *SF3B1*$^{WT}$ (orange). Boxplot median difference = 9.32. $P$ values for **a**–**c** are using two-sided Mann–Whitney $U$ tests. Box-plots show median line, box limits are upper and lower quartile, and whiskers are 1.5× interquartile. **d** *ADTRP* gene isoforms, plotted 5′ to 3′. The 632 bp intron that is differentially included is boxed in purple. A differentially skipped exon is boxed in pink. **e** Percent usage of each isoform in each CLL sample, with colors corresponding to isoforms in **d**. Gray bars represent all other isoforms not plotted in **d**. **f** Top: 3′ splice-site motif of constitutively spliced introns. Bottom: 3′ splice-site motif of significant intron retention events identified from short-read sequencing ($n = 67$).

our focus to only the introns that were significantly differentially retained between *SF3B1*$^{K700E}$ and *SF3B1*$^{WT}$. Using DRIMSeq[54] for testing the IR events we identified from FLAIR isoforms, we found 70 introns were significantly different (corrected $p < 0.1$ and abs(dPSI) >10) with no overlap between these nanopore-identified events and the short-read-identified events. Although there were fewer significant introns found to be downregulated in the mutant (dPSI <−10), the magnitude of downregulation was stronger for those introns (Fig. 4c, Supplementary Data 3). An example of a gene with increased expression of an isoform with increased IR splicing in CLL *SF3B1*$^{K700E}$ is *ADTRP* (androgen dependent TFPI regulating protein), which is involved with blood coagulation[58]. Long reads enabled the identification of coordinated splicing events in *ADTRP*, such as isoforms with a differentially skipped exon coordinated with the differentially retained intron (Fig. 4d, e).

We then looked at splice-site motifs for the differentially retained introns (Fig. 4f and Supplementary Fig. 6). In the IR events identified from the CLL short reads where the IR was more spliced in the mutant *SF3B1* condition, we found a strong TGAC branch point motif[59] 15 bp upstream of the 3′SS (Fig. 4f). While this motif had not been reported before in this context, it was consistent with the position of strong branch point sequences upstream of alternative 3′SSs that were associated with *SF3B1* mutation[22]. Sequence analysis of introns with increased inclusion in *SF3B1*$^{WT}$ identified from both the Nalm-6 and TCGA BRCA short-read data also revealed a TGAC/TGAG motif ~15 bp upstream of the 3′SS (Supplementary Fig. 7d, e), although not at exactly the same position. This further supports an underlying mechanism of *SF3B1*$^{K700E}$ in which the mutant prefers splicing at a 3′SS ~15 bp downstream of a strong branch point[22]. We did not observe the same motifs for the IR events identified from

nanopore sequencing. To further investigate differences between IR events identified from nanopore sequencing compared with short-read sequencing, we looked at the intron length distributions. The median read lengths for nanopore reads were 712–948 bp, suggesting a bias against detecting longer IR. Indeed, the majority of differential IR identified in the nanopore data were under 1,000 bp, much shorter compared with those identified from short reads (Supplementary Fig. 8). Thus, while we were able to identify a strong branch point sequence associated with IRs in several short-read datasets, we were unable to do so in the long reads in part because of a length bias in nanopore reads.

Taking full advantage of the long-read data and using our assembled transcripts and isoform quantification approaches, we identified 143 full-length isoforms that were differentially used (corrected $p < 0.1$ and abs(dPSI) >10) between CLL $SF3B1^{K700E}$ and $SF3B1^{WT}$ (Supplementary Data 4). Of these differentially used isoforms, 53 isoforms contained AS events that were significantly different between mutant and wild type $SF3B1$. Collectively, these isoforms contained 5 alternative 3′SSs, 1 alternative 5′SS, eight retained introns, and 28 exon skipping events that were tested to be significant. Although only a few differentially used isoforms contained more than one significant splicing alteration, nearly all (95.1%) overlapped two or more regions where FLAIR identified multiple splicing choices. Differentially used isoforms in the $LINC01089$, $LINC01480$, $XBP1$ genes contain more IR or partial IR in $SF3B1^{WT}$ relative to $SF3B1^{K700E}$ (Supplementary Fig. 9a–c). These alternative isoforms with retained introns were also coupled with alternative 3′ splicing events, which is a type of regulated coupling that would be difficult to determine confidently with short-read data.

### $SF3B1^{K700E}$ downregulates unproductive intron retention.

Short-read studies have noted an association between mutant $SF3B1$ in CLL and an increase in transcripts with computationally-predicted premature termination codons (PTCs)[19]. With full-length cDNA sequencing, we are given a more accurate representation of the complete transcript and thus are better able to detect transcripts with PTCs and estimate the proportion of unproductive transcripts. Unproductive isoforms are defined as those that have a PTC 55 nucleotides or more upstream of the 3′ most splice junction[60,61] (Fig. 5a). Productive transcripts are presumed to be protein coding, whereas unproductive transcripts are either detained in the nucleus or subject to nonsense-mediated decay (NMD) if exported to the cytoplasm[61–64]. For example, $SRSF1$ has several unproductive transcripts that are known to be either nuclear-retained or NMD-triggering[62], two of which (ENST00000581979.5 and unannotated Isoform V[62]) we were able to identify and accurately predict as unproductive in our nanopore data (Supplementary Fig. 10). We also identified 5 additional unannotated $SRSF1$ isoforms with more than 100 supporting reads, 2 of which are productive and 3 are unproductive.

We found that isoform productivity predicted from local alterations often matched the productivity of the full isoform. That is, for the alternative 3′SSs with more proximal site usage in $SF3B1^{K700E}$, the alterations that were predicted to preserve the reading frame (i.e., those that included a multiple of three basepairs of sequence in the coding region) were more associated with productive isoforms (Table 2). However, there were some exceptions in which alterations that were predicted to disrupt the reading frame were associated with productive isoforms and vice versa, demonstrating the need for long-read data to accurately predict productivity changes caused by alterations at a junction level. Next, we looked systematically at all FLAIR isoforms categorized by their productivity in combination with the presence

or absence of retained introns (Fig. 5b). Although together productive and unproductive IR isoforms were downregulated in $SF3B1^{K700E}$ (Fig. 4a), the reduction was more pronounced in the unproductive IR isoforms (productive-spliced and unproductive-IR two-sided Mann–Whitney $U$ $p = 1.25 \times 10^{-6}$). To further understand the decrease in unproductive IR observed in $SF3B1^{K700E}$, we performed a gene ontology (GO) analysis of the parent genes for the 94 isoforms in that category. No category reached statistical significance (corrected $p < 0.05$); however, the most enriched terms included antigen processing and presentation, cell cycle, regulation of MAP kinase activity, and positive regulation by protein kinase activity (Supplementary Fig. 11 and Supplementary Data 5). The prevalence of cellular signaling GO terms parallels a finding in glioblastoma, where genes with a decrease in detained introns regulated by $PRMT5$ are also associated with perturbed kinase signaling[65].

### Discussion

In this study, we identified splicing changes in the context of full-length isoforms in primary CLL samples with and without a mutation in splicing factor $SF3B1$. We were able to achieve high sequencing depths for long-read sequencing standards using the nanopore PromethION. Across the nine samples with great flow cell to flow cell variability in sequencing depth, we were able to generate 149 million pass reads. The errors in nanopore reads pose a challenge for many existing tools, e.g., alignment artifacts posing as novel splice sites. We developed FLAIR, a tool for the identification of high-confidence full-length isoforms and quantification of AS in noisy long-read data. With FLAIR splice correction using matched CLL short reads, we rescued reads with incorrect splice sites for further analysis. FLAIR then defined a high-confidence isoform set for the nanopore CLL data as follows: (1) the fully corrected reads were collapsed to define a first-pass isoform set with vetted splice junctions, (2) all of the reads were reassigned to an isoform to assist with quantification of the aforementioned isoform set, and (3) isoforms with insufficient support were removed from the isoform set. FLAIR demonstrates improvements over the sparse space of nanopore analysis tools and enabled the discovery of many novel, $SF3B1$ mutant-associated high-confidence isoforms.

Using FLAIR-defined transcripts, we identified aberrant splice site and retained intron usage associated with $SF3B1^{K700E}$. The alternative 3′SS usage patterns were consistent with alterations identified in short-read data. In addition, long-read sequencing highlighted a downregulation of isoforms containing retained introns in $SF3B1^{K700E}$ relative to $SF3B1^{WT}$. This downregulation was corroborated by reanalyzing CLL, Nalm-6 cell line, and TCGA BRCA short-read datasets with mutant $SF3B1$. CLL has been shown to contain elevated levels of splicing alterations, regardless of $SF3B1$ mutation status[16,66]. The subset of introns that exhibited increased splicing in the mutant point to a different intron retention landscape in CLL $SF3B1^{K700E}$. Introns more significantly spliced out in $SF3B1$-mutated samples contained a strong branch point TGAC sequence ~15 bp upstream of the 3′ SS, consistent with previously reported branch site motifs of altered 3′SSs associated with the mutation[22].

Full-length reads also allow for improved identification of IR and classification of transcript productivity, improving our understanding of $SF3B1$ biology in CLL. Most notably, we observed a decrease in expression of intron-retaining isoforms categorized as unproductive in mutant $SF3B1$. Previous publications with short-read sequencing have shown that $SF3B1$ mutation causes lower expression of genes with unproductive isoforms[19]. As short-read sequencing has greater depth, it is easier to detect unproductive transcripts, many of which can be

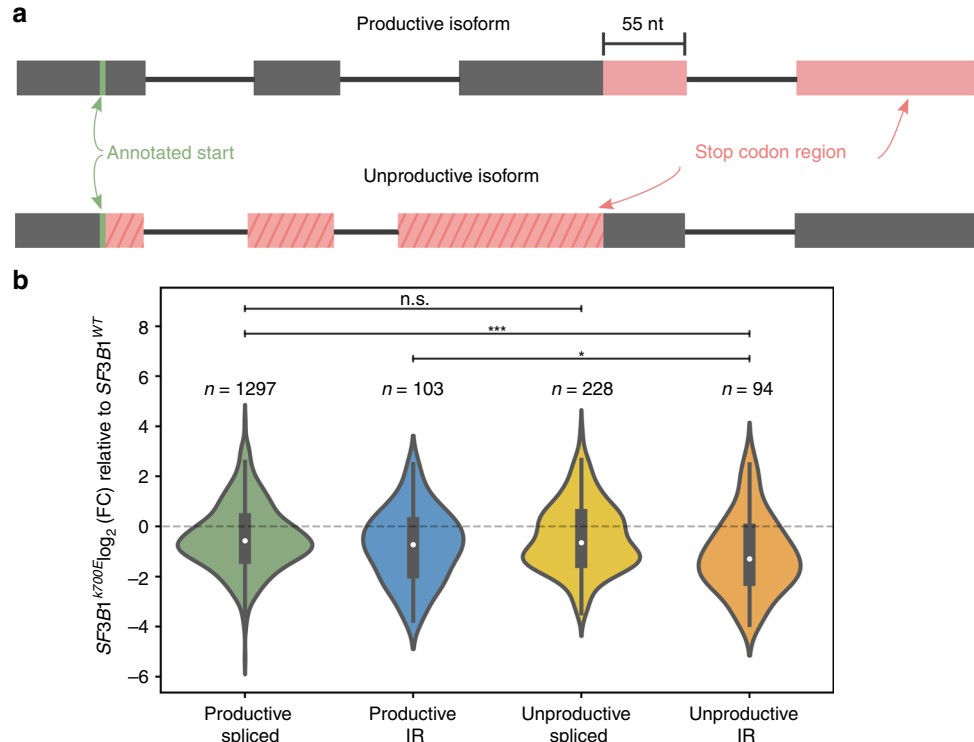

**Fig. 5 Mutant SF3B1 downregulates unproductive, intron-retaining transcripts. a** Schematic of productive and unproductive isoforms. The region on an isoform where stop codons can occur is colored red. Unproductive isoforms have premature stop codons present 55 nt or more from the last splice junction. **b** Expression fold-change (FC) between SF3B1$^{K700E}$ and SF3B1$^{WT}$ of FLAIR isoforms categorized as containing (IR) or not containing (spliced) retained introns and by productivity. The difference between the log2(FC) of the productive spliced and the unproductive-IR categories is 0.720. Violin plot show median as white dot, box limits are upper and lower quartile, and filled area represents the entire range of the kernel density estimation. *$p$ value < 0.05, ***$p$ value < 0.0005, two-sided Mann–Whitney $U$ test. Source data are provided in a Source Data file.

**Table 2 Productivity of isoforms containing significant alternative 3′ events.**

| Alternative 3′ splice-site coordinates | Is a multiple of 3 | Productive isoforms | Unproductive isoforms | Isoforms with no start/stop codon |
|---|---|---|---|---|
| chr2:241335958,241335974 | N | 1 | 1 | 0 |
| chr20:35556954,35556972 | Y | 1 | 0 | 0 |
| chr1:161710439,161710474 | N | 0 | 2 | 0 |
| chr5:133091165,133091192 | Y | 1 | 0 | 0 |
| chr6:15651408,15651363 | Y | 4 | 2 | 0 |
| chr17:81239567,81239594 | Y | 0 | 3 | 1 |
| chr1:1374590,1374445 | N | 0 | 1 | 0 |
| chr12:120498072,120498119 | N | 8 | 0 | 0 |
| chr3:186787478,186787494 | N | 0 | 8 | 0 |
| chr1:155310042,155310063 | Y | 3 | 5 | 0 |
| chr1:45013574,45013593 | N | 1 | 4 | 0 |
| chr2:202291020,202291124 | N | 0 | 8 | 0 |

The alternative 3′ splicing events displayed are those where the proximal splice site is preferred in SF3B1$^{K700E}$ samples. Only splicing events that affect coding regions are shown, and only isoforms using the proximal splice site were counted. Isoforms were counted if they have a minimum of 20 counts across SF3B1$^{K700E}$ samples.

lowly expressed due to NMD. We speculate that the more highly expressed unproductive transcripts we detected with nanopore sequencing are likely retained in the nucleus. Performing a GO analyses revealed kinase signaling associated with the unproductive IR events downregulated in *SF3B1$^{K700E}$*. We postulate that these unproductive-retained introns are the cases of detained introns, as kinase signaling has been associated specifically with detained introns[65]. The downregulation of these unproductive detained introns may result in increased production of kinase-signaling genes to support tumor proliferation. Further experimentation would be necessary to verify that the unproductive IR

events are retained in the nucleus (detained introns), and if there is a functional relationship between kinase-signaling genes.

A subsampling analysis revealed that we have not saturated the number of discoverable isoforms. Despite efforts to obtain nanopore-sequencing data from a more high-throughput-sequencing platform (PromethION) and account for the low accuracy of 1D nanopore sequencing, we note that the read depth, cohort size, splicing complexity, and high error rates in nanopore data are still limiting factors of this study. While we were able to detect alternative 3′SSs recapitulating *SF3B1* biology, nanopore sequencing was not able to detect as many altered events as short-

read sequencing potentially due to the smaller cohort and the difficulty of detecting subtle splicing alterations. The small overlap between nanopore-identified and short read-identified alternative 3′SSs could also be due to the stringent filtering applied in an effort to determine AS more accurately. In addition, we did not find a strong branch point motif near the 3′SS of nanopore-identified IR events. This may have been due to a smaller cohort size and the bias toward shorter retained introns (<1000 bp) sequenced in long reads (Supplementary Fig. 8). Everything considered, studying splicing factor mutations in primary samples using nanopore sequencing with fewer reads than the current study or without short-read sequencing would be suboptimal. Short-read sequencing was necessary for increasing confidence in splice sites, although future work with higher-accuracy reads could potentially obviate the need for short reads. Future studies of primary samples should also include larger cohort sizes, with three samples per genotype being the minimum[67]. Even though short-read technology is able to sequence more deeply than long reads, the ability of short reads to saturate splice junction detection is depth-dependent[68]; thus, splicing studies should aim to sequence as deeply as possible. Fortunately, the throughput and accuracy of nanopore and PacBio technology has the potential to increase with subsequent iterations of the technologies[69]. For nanopore in particular, methods to achieve higher sequence accuracy[37] or circumvent PCR bias and reverse transcription length restrictions[30] have been developed. In line with the rigorous pace of improvements in the field of long reads, PacBio has recently improved their throughput 8× with the newest PacBio Sequel II system, which has been shown to generate ~19 and 83 Gb of consensus reads[39,70].

This study of six primary CLL samples with nanopore sequencing demonstrates the ability of the nanopore to identify and quantify cancer-specific transcript variants. Long reads enabled us to better identify IR events, better estimate isoform productivity, and observe AS complexity in full-length isoforms. Ultimately, nanopore sequencing facilitated the building of a more complete picture of the transcriptome in primary cancer samples. With the impending rapid growth of long-read sequencing, tools like FLAIR will be useful in identifying key disease-associated variants that may serve as biomarkers of potential prognostic or therapeutic relevance.

## Methods

**Data generation and handling.** Peripheral blood mononuclear cells were obtained from patients with CLL and from healthy adult volunteers, enrolled on sample collection protocols at Dana-Farber Cancer Institute, approved by and conducted in accordance with the principles of the Declaration of Helsinki and with the approval of the Institutional Review Board (IRB) of Dana-Farber Cancer Institute. Samples were cryopreserved in 10% DMSO until the time of RNA extraction. RNA was extracted from tumor or normal samples using the Qiagen RNeasy Mini Kit[17]. The sample IDs of the CLL $SF3B1^{WT}$ samples are CW67 (WT 1), CW95 (WT 3), and JGG0035 (WT 2) and the IDs of the $SF3B1^{K700E}$ samples are DFCI-5067 (MT 1), CLL043/CW109 (MT 2), and CLL032/CW84 (MT 3) from Wang et al.[17] JGG035 is the only sample not included in that study. All samples had RIN scores above 7. The extracted RNA was reverse transcribed using the SmartSeq protocol[71] and cDNA was PCR-amplified using KAPA Hifi Readymix 2× (98 °C for 20 s, 67 °C for 16 s, 72 °C for 4 min)[34]. Fifteen cycles of PCR were performed. Prior to library preparation, the concentration of the cDNA for the samples ranged from 1.26 to 10.7 ng/μl (Supplementary Table 1). Oxford MinION 2D amplicon libraries were generated according to the Nanopore community protocol using library preparation kit SQK-LSK208 and sequenced on R9 flowcells. Basecalling was performed with albacore v1.1.0 2D basecalling using the—flow cell FLO-MIN107 and—kit SQK-LSK208 options. The same cDNA preparation protocol was used for PromethION sequencing. Library preparation for 1D sequencing was performed following Oxford Nanopore's protocol with the exception of the last bead clean up using a 0.8× bead ratio. The PromethION libraries were prepared and sequenced in one batch of 3 and one batch of 6, with at least 1 sample of each condition in each batch (Supplementary Table 2). Basecalling of 1D PromethION reads was done with guppy v2.3.5 with the default options, and only reads that were designated pass reads in the summary file were used for subsequent analyses. We identified reads with adapter sequences on both ends following the approach employed in the

MandalorION pipeline:[34] (1) adapters are aligned to all the reads using blat[72], (2) if there are at least ten bases at the left and right ends of the reads that match the adapter sequence then the read is considered to have adapters on both ends. We found that only a fraction of our reads that were called as pass reads contained the adapter sequences on both ends (~35–55%). In the interest of being able to use more of our data, we did not remove these reads from the analyses.

**Nanopore-sequencing statistics.** The reads for each sample were aligned with minimap2 v2.7-r654[44] to the GENCODE v24 transcriptome and the read-isoform assignments were determined using the primarily alignments. Following read-isoform assignment, the percentage of full-length reads was calculated as the number of reads covering 80% of nucleotides for the transcript they were assigned to divided by the total number of reads that aligned. The number of genes observed was computed by counting the genes represented by all the isoforms. Genes with multiple isoforms identified were considered alternatively spliced.

**Spliced alignment and read correction.** Reads were aligned to hg38 (http://hgdownload.cse.ucsc.edu/goldenPath/hg38/bigZips/) using minimap2 v2.7-r654[44] in spliced alignment mode with the command 'minimap2 -ax splice'. GMAP 2017-10-30[45] was used for comparison against minimap2. Indels were removed from the read alignments. FLAIR *correct* (v1.4) was used to correct the splice-site boundaries of reads. All splice sites were assessed for validity by checking for support in GENCODE v24 comprehensive annotations or short reads. Splice junctions were extracted from matched short-read data, and only the junctions supported by three uniquely mapping short reads were considered valid. Incorrect splice sites were replaced with the nearest valid splice site within a 10-nt window. The set of corrected reads consists of reads that contain only valid splice sites.

**Data simulation.** We created a wrapper script for NanoSim[49] to generate simulated transcriptomic data. We simulated reads from transcripts of genes that were found to be expressed in our real nanopore data, with read lengths modeled after real data using our wrapper script (see "Code availability"). We used the hg38 nanopore error model simulated from step 1 of NanoSim[49].

**Isoform-identification methods.** We used Mandalorion version II[34] and FLAIR v1.4 to assemble isoforms on the SIRV and simulated data with tuned read support threshold parameters. FLAIR was run with the default settings. Mandalorion was run with the parameters on the GitHub '-u 5 -d 30 -s 200 -r 0.05 -i 0 -I 100 -t 0 -T 60' varying -R (minimum number of reads for an isoform to be reported). Consensus isoform sequences were aligned to the genome with minimap2 and converted to gtf prior to running GFFCompare[48]. Pinfish was run with default parameters, adjusting the minimum cluster size parameter only. For running FLAIR on the PromethION CLL/B cell data, the following FLAIR *collapse* algorithm was followed: to assemble the first-pass assembly, transcription start sites and transcription end sites are determined by the density of the read start and end coordinates. We compared 100-nt windows of end sites and picked the most frequently represented site in each window (-n best_only). The final nanopore-specific reference isoform assembly is made by aligning raw reads to the first-pass assembly transcript sequence using minimap2, keeping only the first-pass isoforms with a minimum number of three supporting reads with MAPQ ≥ 1. All pass reads, including reads that did not contain sequenced adapters on both ends, were used when running FLAIR as FLAIR is equipped to deal with truncated reads; information can be gleaned from truncated reads of sufficient length to be assigned to an isoform and the reads that are too short for a unique assignment are excluded from the isoform quantification.

**Saturation analysis.** We performed the saturation analysis on the three runs with the most coverage in each sample (WT 3, MT 2, B Cell 1). The total reads from each run was used, in addition to subsampled sets of reads. Reads were subsampled in increments of 10 million by random selection using python random.sample(). We used FLAIR to identify isoforms within each subset of reads using the 'best_only' parameter to obtain only one transcription start and end site per splice junction chain.

**Isoform quantification and fold-change calculation.** Isoforms were quantified using FLAIR *quantify*, only counting the alignments with quality scores of 1 or greater. Isoform counts within each sample were normalized by dividing each count by the upper quartile (75th percentile) of the read counts of protein-coding genes. Only genes labeled as protein coding in GENCODE v24 annotation were considered protein coding.

**AS event calling and statistical testing.** Custom scripts were written for FLAIR to identify alternative acceptor, alternative donor, cassette exon skipping, and intron retention events (FLAIR *diffSplice*). Alternative 3′SS were grouped by the 5′ SS they were observed with and had to be present in overlapping exons and vice versa for alternative 5′SS calling. Alternative 3′ and 5′ splice sites that were within 10 bp of each other were exempt from statistical tests. For the analysis of the pilot data containing one wild type and one mutant $SF3B1$, a Fisher's exact test was used

to determine the significance of AS events. For analysis of the PromethION data with replicates, we used DRIMSeq[54]. DRIMSeq statistical testing accounted for sequencing and RNA batch according to batch numbers for each sample (Supplementary Table 1). The expression filters used for DRIMSeq were as follows: a minimum of 4 of the 6 samples should cover either the inclusion or exclusion event with minimum coverage of 25 reads. Of the 4 samples with sufficient coverage, 2 should be from either the CLL *SF3B1* WT or *SF3B1* K700E condition. A pseudocount of 1 was used to prevent events with 1–2 dropout samples from being excluded from testing. For differential isoform usage testing (FLAIR *diffExp*), isoforms were grouped by gene and only genes with at least 25 reads in 4 of the 6 samples were tested. As these libraries were poly(A) selected, we did not distinguish between IR due to incomplete transcript processing and IR deliberately retained due to sample genotype.

**Intron retention and productivity analysis.** Fold-change was calculated using the median upper-quartile-normalized isoform count for each condition and dividing the mutant expression by the wild-type expression. Only the transcripts with a median of ten or more in one of the conditions were plotted. IR were defined as any intron that is completely spanned by another isoform's exon. For identification of NMD-sensitive transcripts, we used annotated start codons from GENCODE v24 and translated the full-length assembled isoforms. Isoforms with a PTC were called unproductive, and isoforms without PTCs were called productive. A PTC was defined as a stop codon detected before 55 nucleotides or more upstream of the last splice junction[60]. If a transcript overlapped more than one annotated start codon, the productivity was assessed by using (1) the 5′ most start codon or (2) the start codon yielding the longest transcript; if both strategies yielded different productivity results, then the isoform was excluded from analysis.

**GO analysis.** GO analysis was performed with the R package goseq v1.32.0[73], setting the parameter method = hypergeometric to remove the correction for gene length bias that affects short-read data. GO terms with only one term in the category were removed from further analysis.

**Reporting summary.** Further information on research design is available in the Nature Research Reporting Summary linked to this article.

## Data availability

A reporting summary for this Article is available as a Supplementary Information file. Normal B cell and CLL nanopore signal data will be deposited in the database of Genotypes and Phenotypes (dbGAP: phs001959.v1.p1). The source data underlying Figs. 3b, 4a, b, 5b are provided as a Source Data file. All data is available from the corresponding author upon reasonable request.

## Code availability

Scripts developed from this work can be found at https://github.com/BrooksLabUCSC/FLAIR. These scripts include the wrapper script flair.py, which can be used to run the FLAIR modules. The NanoSim_Wrapper.py script, a wrapper for simulating transcript reads using the nanopore genomic read simulator Nanosim, can be found at https://github.com/BrooksLabUCSC/labtools.

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

## Acknowledgements

We acknowledge Wandi Zhang for locating and facilitating the delivery of the CLL and B-cell RNA samples. We also would like to acknowledge Miten Jain, Hugh Olsen, Ashley Byrne, and Christopher Vollmers for their guidance with nanopore library preparation and sequencing and Miten Jain for assistance basecalling. Finally, we acknowledge Winston Timp, Mark Akeson, Hugh Olsen, Miten Jain, Rachael Workman, Roham Razaghi, and Timothy Gilpatrick for comments on FLAIR. This work was supported by the Damon Runyon Cancer Research Foundation and the Pew Charitable Trusts to A.N. B. A.D.T. was funded through NIH grant 5T32HG008345. A.D.T was partially supported by R01HG010053 (PI: Mark Akeson, UC Santa Cruz). C.M.S. was supported by training grants NIH T32GM008646, 2R25GM058903, and the Ford Foundation predoctoral fellowship. C.J.W. is a Scholar of the Leukemia and Lymphoma Society (LLS) and acknowledges support from NCI (1RO1CA155010-01A1; 1U10CA180861-01; 1P01CA206978-01).

## Author contributions

A.D.T. and A.N.B. designed the study. A.D.T. performed experiments. A.D.T., C.M.S., M.J.B., K.H., A.N.B. wrote code and analyzed data. A.D.T., C.M.S., M.J.B., K.H., E.H.R., C.J.W., A.N.B. interpreted the data. C.J.W. provided CLL samples and donor samples. A.D.T. and A.N.B. wrote the manuscript with input from all other co-authors.

## Competing interests

A.D.T. and A.N.B. have been reimbursed for travel, accommodation, and registration for conference sessions organized by Oxford Nanopore Technologies. C.J.W. is a co-founder of Neon Therapeutics, Inc and is a member of its scientific advisory board, and receives research funding from Pharmacyclics. All other authors have declared no competing interests.
