## [Peer Review File · Nature Communications]

Reviewers' comments:

Reviewer #1 (Remarks to the Author):

GENERAL COMMENTS

The aim of this study is to characterize transcriptional changes associated with SF3B1 mutation in CLL using nanopore long-read sequencing. SF3B1 is a splicing factor with hotspot mutations in multiple cancers, and is one of the most frequently mutated genes in CLL. Characterizing alterations associated with SF3B1 mutation is of great interest, as it is assumed that they can explain some of the selective advantage conferred by the mutation. Alterations have been extensively analyzed using short-read RNA-seq data, but new long-read sequencing technologies have the potential to provide a more complete characterization. The authors describe a new computational workflow FLAIR (Full-Length Alternative Isoform analysis of RNA) for isoform analysis from long-read data. They generate nanopore data from three samples, two CLL samples with and without SF3B1 mutation, and one normal B-cell sample. Identified transcript alterations associated with SF3B1 mutations confirm alternative splice site usage, previously identified from short-read data. In addition the authors report on differences in intron retention between the SF3B1 mutant and wild-type CLL samples. They observe increased intron retention in the SF3B1 wild-type sample, and normal (B-cell-like) intron retention levels in the SF3B1 mutant sample. The authors take advantage of full-length transcript characterizations to predict functional consequences of splice alterations.

SPECIFIC COMMENTS

- 1) Low read depth and small cohort size are significant limitations. Only two CLL samples and one normal B cell sample are sequenced to obtain < 300K reads in total. Can the samples be resequenced to achieve higher read depth? Data from additional CLL samples should be generated to corroborate findings.
- 2) p-values testing for differential splice events and isoform usage should be corrected for multiple testing to avoid spurious calls.
- 3) Only 11 alternative 5'/3' splice site alterations and 25 intron retention events could be identified from the nanopore data. What is the overlap between events identified from long-read data and short-read data (65 A3SS, 67 IR)? It is not entirely clear what new insights are gained from the nanopore data compared to previous short-read data (e.g. the intron retention analysis in Fig. 4C is based on events identified from short-read data).
- 4) For the 122 differential isoforms, please include a breakdown in terms of the type of splice alterations e.g. A3SS, A5SS, SE, IR etc.
- 5) Claims regarding global down regulation of IR transcripts seem overstated as the differences shown in Fig 4A/B are very subtle. The text should mention effect size as well as p-values.
- 6) Since the manuscript focuses on intron retention changes, these could be followed up by reanalyzing data from controlled experiments, e.g. data from isogenic cell lines with and without SF3B1 mutation (Darman et al, Cell Reports 2015) or data from over-expression studies (Wang et al. Cancer Cell 2016). Is reduced intron retention specific to CLL or also observed for other cancers with hotspot mutations in SF3B1?
- 7) For the last section, while some unproductive transcripts like for SRSF1 may be retained in the nucleus, it is not clear why the authors conclude that unproductive transcripts are generally retained - cf. "The majority of isoforms (68%) with intron retention events were determined to be unproductive, suggesting that these are detained introns that are localized to the nucleus." Also

the interpretation of observed differences between productive and unproductive IR transcripts seems speculative and should be moved to discussion.

Reviewer #2 (Remarks to the Author):

The authors present a new workflow called FLAIR (Full-Length Alternative Isoform analysis of RNA) that facilitates analysis of full-length transcript isoforms using nanopore sequencing data. This offers much better opportunities to characterize exon connectivity and intron retention that are difficult to study unambiguously with short reads. They use this workflow to reanalyze transcriptome data from one CLL SF3B1WT patient, one CLL SF3B1K700E patient, and one normal B cell sample. The FLAIR pipeline works very well to create consensus isoforms from reads with high error rates, and the main findings about splicing seem to agree in general with results obtained from short read data. The splicing community would benefit from a good study like this that investigates methods and parameters for analysis of nanopore data.

1. The FLAIR pipeline does a good job of defining splice junctions and assembling full length isoforms from nanopore data with an inherently high error rate.
2. To my knowledge the discovery of intron retention differences in SF3B1K770E CLL patients and SF3B1WT CLL patients is novel and exciting.
3. p. 5: The statement that intron retention is globally down-regulated in the mutant CLL sample compared to CLL SF3B1WT is intriguing but would benefit from some clarification. Does the term "global" imply that the majority of retained introns differ between these CLL classifications, or is this result driven by the 25 differentially retained introns mentioned at the end of the paragraph? What is the actual difference in percent spliced in for retained introns between these groups? Presumably in absolute terms it's a small difference. Finally, how much overlap is there between the groups of differentially retained introns identified with nanopore data (25 introns) vs short read data (67 introns)?
4. One of the most interesting questions is, does the long read data provide insights into intron retention differences in SF3B1K700E vs SF3B1WT that could not be deduced from short read data? Logically the interpretation of intron data will be more solid in principle from long read data, but in practice how much of a difference is there?
5. Do the authors care to speculate on some guidelines for use of nanopore sequencing, e.g., how many reads are necessary to achieve good isoform predictions for moderately expressed genes if one does not have short read data to aid interpretations? Or is nanopore not ready to stand on its own yet for splicing studies, due to difficulties in obtaining sufficient reads from multiple biological replicates?
6. Related to #5: Obviously acquiring this pioneering data was not easy, and applying it to human disease sample for which comparable short read data is available is highly commendable. The question remains as to how reliable is the nanopore data based on one biological replicate in each group, if one did not have supporting short read data?

Minor issues:

1. For readers not familiar with CLL, it would be nice to explain why the normal B cell transcriptome is an appropriate control in this study.
2. The first paragraph of the discussion reads as if these conclusions were deduced from nanopore data alone, but actually the analysis relied on a combination of nanopore and short read data. This impression should be corrected.

3. Figure 2A/2B: what is the difference between the splice site accuracy information in the blue minimap data in 2A, vs the minimap data shown in 2B? The corrected data in 2B resembles minimap data in 2A, but it's not identical.
4. Figure S5A: what is the difference between the gray and green isoforms? Fig. S5D: What is the difference between the gray isoform and the blue isoform? Are there differences in transcription start and stop sites?
5. Abstract: In the text "no difference between CLL SF3B1MT and B cell", is there a mistake in the superscript, i.e., should it be "CLL SF3B1WT"? If not, please explain SF3B1MT.

Reviewer #3 (Remarks to the Author):

The manuscript by Tang et al describes FLAIR; a tool developed to analyze full length transcripts including detection and quantification of alternative isoforms from error prone long read sequencing data. To demonstrate the use of FLAIR, cDNA was generated in SF3B1 wild type and mutant chronic lymphocytic leukemia primary cells and long read sequencing data was acquired using Oxford Nanopore Technologies MinION sequencing. The paper presents what appears to be an extremely useful new tool and demonstrates its efficacy in a compelling biological context

1. There is less technical detail on the sequencing in the main paper than might be ideal. For example the 'Nanopore Sequencing Statistics' section, it is unclear if this data was generated from a single or multiple flowcells. The amount of sequencing generated is not detailed. And the gene coverage appears lower than might be anticipated. In addition, as the authors mentioned in the discussion, multiple runs may create a batch effect. Additionally, the greater than 2X difference in 2D pass reads between B cell and CLL should be addressed.
2. The key improvements vs other tools/approaches are mentioned but not explored in much detail. This seems like a fundamental part of the paper so it was surprising to have to read the results section several times to make sure I didn't miss more exposition. Figure 2C provides more detail than the text and the tool looks good in these comparisons. However, I wasn't clear why coverage was displayed on the x axis since it seemed to contribute minimally to precision/recall. Finally, I was confused by the use of 'increased' when describing the sensitivity of the new tool ("higher" maybe?).
3. Isoform usage (fig 4d and Supplemental 5): These figures represent FLAIR's ability to quantify isoforms in specific genes allowing for proportional analysis. However, the low number of isoform reads for each gene makes the reproducibility of these plots questionable. Replicating the experiment would increase this confidence.
4. Methods: PCR was used to add adapters to the cDNA. As PCR has the potential to introduce bias in quantification of the isoforms, the methods should explicitly state PCR parameters including amplification cycles.
5. Short-read validation: The authors state "removed any novel, nanopore-only splice sites from our analysis due to limited evidence of their validity." This seems to suggest the authors believe short reads are necessary to validate the findings of the long reads. However, it was stated in the manuscript that FLAIR can be used with or without the short read data. The utility of FLAIR without needing short reads would seem to be a significant advancement over current tools/methods. Can the authors clarify this and potentially add data to back up their position.
6. Additional information: A table containing Number of genes sequenced, coverage, Gb of data, median length, number of genes with differential splicing etc. would be interesting and useful to help future users of FLAIR to decide to use a global or targeted RNAseq approach.
7. The way the aberrant splicing landscape is described was confusing to me: in some cases, there seems to be the suggestion that the wild type cells have "aberrant" intron retention? Can you clarify this for the general reader.

8. It is unclear how much the GO analysis adds to the manuscript as it is currently presented.

Minor:

1. Abstract: '...and no difference between CLL SF3B1MT'. MT—we assume this should be WT?
2. Results: Flair Provides improved full-length isoform detection....: Figure 3C is referenced, however in this context 2C should have been referenced?
3. Figure 4a: The P-value indicates significance, however displayed like this, the fold-change difference appears very small.
4. Early clarification that a global RNAseq approach was taken should be considered for ease of understanding.
5. Productive vs unproductive terminology. Can you provide further bibliographical support for this construct?
6. 'Sites that align outside of the wiggle room'. Please define wiggle room
7. 'which we resolve by smoothing gaps of 30 nt or fewer' please clarify how this was accomplished.
8. Statistical tests are quoted with p values but the comparator groups are hard to discern.
9. FLAIR workflow great and helps orient the reader. A similar panel demonstrating the experimental workflow may help the reader navigate the manuscript.
10. Generally the figure legends could use more detail allowing the reader to understand the figure alone without flipping back and forward to the text multiple times.

Reviewer #1 (Remarks to the Author):

GENERAL COMMENTS

The aim of this study is to characterize transcriptional changes associated with SF3B1 mutation in CLL using nanopore long-read sequencing. SF3B1 is a splicing factor with hotspot mutations in multiple cancers, and is one of the most frequently mutated genes in CLL. Characterizing alterations associated with SF3B1 mutation is of great interest, as it is assumed that they can explain some of the selective advantage conferred by the mutation. Alterations have been extensively analyzed using short-read RNA-seq data, but new long-read sequencing technologies have the potential to provide a more complete characterization. The authors describe a new computational workflow FLAIR (Full-Length Alternative Isoform analysis of RNA) for isoform analysis from long-read data. They generate nanopore data from three samples, two CLL samples with and without SF3B1 mutation, and one normal B-cell sample. Identified transcript alterations associated with SF3B1 mutations confirm alternative splice site usage, previously identified from short-read data. In addition the authors report on differences in intron retention between the SF3B1 mutant and wild-type CLL samples. They observe increased intron retention in the SF3B1 wild-type sample, and normal (B-cell-like) intron retention levels in the SF3B1 mutant sample. The authors take advantage of full-length transcript characterizations to predict functional consequences of splice alterations.

We thank the referee for the accurate summary of our paper.

SPECIFIC COMMENTS

1) Low read depth and small cohort size are significant limitations. Only two CLL samples and one normal B cell sample are sequenced to obtain < 300K reads in total. Can the samples be resequenced to achieve higher read depth? Data from additional CLL samples should be generated to corroborate findings.

We have resequenced the same 3 samples using ONT PromethION, which is a platform that yields the highest long-read throughput from a single run. In addition, we have sequenced 4 other CLL samples, 3 of which are from Wang et al., Cancer Cell 2016 and 2 additional B cell samples. In total, we have added PromethION sequencing data for 3 CLL *SF3B1*^{WT}, 3 CLL *SF3B1*^{K700E}, and 3 B cells totaling 149 million pass reads, increasing the amount of data from the original submission (<1 million reads) by several orders of magnitude.

2) p-values testing for differential splice events and isoform usage should be corrected for multiple testing to avoid spurious calls.

We have updated all the existing analyses (alternative 3' SS, differential isoform usage, differentially retained introns) using the promethION data and all p-values have had multiple testing corrections applied.

3) Only 11 alternative 5'/3' splice site alterations and 25 intron retention events could be identified from the nanopore data. What is the overlap between events identified from long-read data and short-read data (65 A3SS, 67 IR)? It is not entirely clear what new insights are gained from the nanopore data compared to previous short-read data (e.g. the intron retention analysis in Fig. 4C is based on events identified from short-read data).

We have noted the number of overlapping significant A3SS and IR events as identified by nanopore and short read in the text (2 A3SS, 0 IR). Although the same short read-identified A3SS were not identified as significant in our nanopore analysis, the Pearson correlation of delta PSIs for high-coverage events from our nanopore data versus the delta PSIs of the same splice sites quantified by short read data was 0.952 (**Figure 3**). As for novel insights gained, nanopore sequencing offers long range splicing information. **Figure 3C** shows an alternative 3' SS identified from our analysis of nanopore data and the diversity of isoforms expressing that A3SS. In the Discussion section, we highlight the additional insights that are gained with long-read data:

p. 10: “ Long reads enabled us to better identify IR events, better estimate isoform productivity, and observe AS complexity in full-length isoforms. Ultimately, nanopore sequencing facilitated the building of a more complete picture of the transcriptome in primary cancer samples. “

4) For the 122 differential isoforms, please include a breakdown in terms of the type of splice alterations e.g. A3SS, A5SS, SE, IR etc.

The updated analysis, including replicates, finds 143 differential isoforms. To identify different classifications of alternative splicing types from long-read data, we added a new module called *diffSplice* to FLAIR which performs these classifications. We have added a breakdown of the number of each splicing alteration found in all the differential isoforms into the text:

p. 8: “Collectively, these isoforms contained 5 alternative 3' SSs, 1 alternative 5' SS, 8 retained introns, and 28 exon skipping events that were tested to be significant.”

5) Claims regarding global down regulation of IR transcripts seem overstated as the differences shown in Fig 4A/B are very subtle. The text should mention effect size as well as p-values.

With the added statistical power with increased samples and sequencing depth, the downregulation of IR is more pronounced. We have included in the legend what the difference between the medians of both groups is (in this case, $0.395 \log_2 FC = 1.3$ fold change). When looking at the significantly downregulated IR events, the median delta PSI is ~25 (Fig. 4C).

6) Since the manuscript focuses on intron retention changes, these could be followed up by reanalyzing data from controlled experiments, e.g. data from isogenic cell lines with and without SF3B1 mutation (Darman et al, Cell Reports 2015) or data from over-expression studies (Wang et al. Cancer Cell 2016). Is reduced intron retention specific to CLL or also observed for other cancers with hotspot mutations in SF3B1?

We obtained the short-read data isogenic Nalm-6 cell lines from Darman et al., Cancer Reports 2015, which contains 3 SF3B1 WT and 3 SF3B1 K700E samples. We have noted in Supplemental Figure 7 the change in IR as detected by JuncBASE, a short-read alternative splicing quantification software. It appears there is a trend for reduced global retention in the SF3B1 K700E data, although the change was not significant according to a Kruskal-Wallis test of differences. The K700E-downregulated IR events also have higher delta PSIs (Supplemental Figure 7B), similar to the nanopore data (Fig. 4C). In addition, we looked at SF3B1-associated splicing in TCGA BRCA JuncBASE samples. For this dataset, we observe significant downregulation of IR in the K700E samples as well.

7) For the last section, while some unproductive transcripts like for SRSF1 may be retained in the nucleus, it is not clear why the authors conclude that unproductive transcripts are generally retained - cf. "The majority of isoforms (68%) with intron retention events were determined to be unproductive, suggesting that these are detained introns that are localized to the nucleus." Also the interpretation of observed differences between productive and unproductive IR transcripts seems speculative and should be moved to discussion.

The interpretation has been moved to and rewritten in the discussion section:

p. 10: "Performing a gene ontology analyses revealed kinase signaling associated with the unproductive intron retention events downregulated in SF3B1^{K700E}. We postulate that these unproductive intron retentions are cases of detained introns, as kinase signaling has been associated specifically with detained introns of isoforms that are localized in the nucleus. The downregulation of these detained introns may result in increased production of kinase signaling genes to support tumor proliferation. Further experimentation would be necessary to verify that the unproductive intron retention events are retained in the nucleus (detained introns) and if there is a functional relationship between kinase signaling genes."

Reviewer #2 (Remarks to the Author):

The authors present a new workflow called FLAIR (Full-Length Alternative Isoform analysis of RNA) that facilitates analysis of full-length transcript isoforms using nanopore sequencing data. This offers much better opportunities to characterize exon connectivity and intron retention that are difficult to study unambiguously with short reads. They use this workflow to reanalyze transcriptome data from one CLL SF3B1WT patient, one CLL SF3B1K700E patient, and one normal B cell sample. The FLAIR pipeline works very well to create consensus isoforms from reads with high error rates, and the main findings about splicing seem to agree in general with results obtained from short read data. The splicing community would benefit from a good study like this that investigates methods and parameters for analysis of nanopore data.

We thank the referee for recognizing the contribution of this study to the broader community.

1. The FLAIR pipeline does a good job of defining splice junctions and assembling full length isoforms from nanopore data with an inherently high error rate.

We thank the referee for recognizing the utility of FLAIR. We have continued to update FLAIR in the past couple months and will continue to update FLAIR to improve user-friendliness and speed to lower the barrier for adoption.

2. To my knowledge the discovery of intron retention differences in SF3B1K770E CLL patients and SF3B1WT CLL patients is novel and exciting.

We thank the referee for appreciating the discovery. Previous CLL SF3B1 splicing publications (e.g., Wang et al. 2008) have shown that there are many intron retention differences between SF3B1 K700E/WT, following alternative 3' SS in prevalence; however, these types of events have not been highlighted or followed up on in more detail. For example, a motif associated with these IR events have not been previously reported. Moreover, long-read sequencing allows for unambiguous detection of intron retention as opposed to short-read sequencing.

3. p. 5: The statement that intron retention is globally down-regulated in the mutant CLL sample compared to CLL SF3B1WT is intriguing but would benefit from some clarification. Does the term "global" imply that the majority of retained introns differ between these CLL classifications, or is this result driven by the 25 differentially retained introns mentioned at the end of the paragraph? What is the actual difference in percent spliced in for retained introns between these groups? Presumably in absolute terms it's a small difference. Finally, how much overlap is there between the groups of differentially retained introns identified with nanopore data (25 introns) vs short read data (67 introns)?

By 'global' we mean from all the isoforms assembled by FLAIR that contain a retained intron. In the text, we specify which analyses are global (investigate all IR) versus the analyses which only focus on those that are significantly differentially retained.

We have updated the IR analysis to include our 3 CLL SF3B1 K700E and SF3B1 WT replicates. We identify 77 significantly differentially retained introns. The magnitude of dPSI between IR that is more retained in SF3B1 K700E samples or more retained in SF3B1 WT is significant and shown in Fig. 4c. There is a larger magnitude of dPSI in those introns more retained in SF3B1 WT. There are no overlapping intron retention events between nanopore and short read data. As we now note in the results and discussion, this may be due to a limitation of long-read technology where the reads are not long enough to sequence transcripts with longer retained introns (>100bp).

4. One of the most interesting questions is, does the long read data provide insights into intron retention differences in SF3B1K700E vs SF3B1WT that could not be deduced from short read

data? Logically the interpretation of intron data will be more solid in principle from long read data, but in practice how much of a difference is there?

As you mention, there is more confidence in the identification of intron retention from long-read data as the read spans the entire intron retention event. Unfortunately, as noted in our new analysis, we find a length bias in the IR events identified in our data because the read lengths are not long enough. A major benefit to performing long-read analysis is the ability to see coordinated splicing with differential intron retention since IR events identified with nanopore reads are determined from full transcripts. This is particularly important for looking at splicing factor mutations as splicing complexity is high (e.g. Fig. 4d-e, Supplementary Figure 9).

5. Do the authors care to speculate on some guidelines for use of nanopore sequencing, e.g., how many reads are necessary to achieve good isoform predictions for moderately expressed genes if one does not have short read data to aid interpretations? Or is nanopore not ready to stand on its own yet for splicing studies, due to difficulties in obtaining sufficient reads from multiple biological replicates?

We think short-read data is necessary for studying splicing factor mutations. Even for a well-annotated genome such as human, it is difficult to make a recommendation on whether nanopore is sufficient to stand on its own, without short-read data.

We performed a saturation analysis (Supplementary Figure 4) to show that even with the depth we have achieved now, particularly with the mutant *SF3B1* sample, we have not saturated isoform discovery. However, similar analyses from short-read studies have shown that even short-read sequencing does not provide the power to discover all variants (Pan et al. 2008; Deveson et al. 2018), and also recommend future studies to sequence more deeply.

The throughput of nanopore technology has increased greatly; the minION, and if available the promethION, have higher yields than when the pilot samples were sequenced. We now have 149 million reads for 9 samples -- an ample amount of reads covering 40k+ genes (Table 1).

To summarize this, we have added the following to the discussion:

p. 10: "studying splicing factor mutations in primary patient samples using nanopore sequencing with fewer reads than the current study or without short-read sequencing would be suboptimal. Future studies of primary samples should also include larger cohort sizes. Fortunately, the throughput and accuracy of nanopore technology continues to increase with subsequent iterations of the technology and methods to achieve higher sequence accuracy or circumvent reverse transcription length restrictions have been developed."

6. Related to #5: Obviously acquiring this pioneering data was not easy, and applying it to human disease sample for which comparable short read data is available is highly commendable. The question remains as to how reliable is the nanopore data based on one biological replicate in each group, if one did not have supporting short read data?

We would not expect any data, Illumina included, to be reliable with one biological replicate in each group. We appreciate that the reviewer recognizes the difficulty in obtaining the initial data, as well as the new data. Regardless, as also suggested by Reviewer #1, we have now included 3 replicates per condition which increases confidence in the conclusions drawn from the data.

Minor issues:

1. For readers not familiar with CLL, it would be nice to explain why the normal B cell transcriptome is an appropriate control in this study.

This has been added to the main text, page 3:

“We also resequenced three normal B cell samples, which are the normal lineage cellular complement to CLL, to use as a normal tissue control for CLL.”

2. The first paragraph of the discussion reads as if these conclusions were deduced from nanopore data alone, but actually the analysis relied on a combination of nanopore and short read data. This impression should be corrected.

The discussion has been reworded to address this. We state the role of short reads in correcting spurious splice junctions from nanopore reads (“With FLAIR splice correction using matched CLL short reads, we rescued reads with incorrect splice sites for further analysis”) in the first paragraph of the discussion. Then, we state the findings from the long-read isoforms (“Using FLAIR-defined transcripts, we identified aberrant splice site and retained intron usage associated with *SF3B1*^{K700E}...”) in the second paragraph.

3. Figure 2A/2B: what is the difference between the splice site accuracy information in the blue minimap data in 2A, vs the minimap data shown in 2B? The corrected data in 2B resembles minimap data in 2A, but it’s not identical.

To reduce confusion, we have merged 2a and 2b into a single panel (Fig. 2a). There is a gmap histogram (yellow), overlaid onto two minimap2 histograms, one with FLAIR splice correction (green) and the other without (blue). In the previous submission, the uncorrected data in 2B and the minimap2 distribution in 2A were both shown in blue and were identical distributions.

4. Figure S5A: what is the difference between the gray and green isoforms? Fig. S5D: What is the difference between the gray isoform and the blue isoform? Are there differences in transcription start and stop sites?

Figure S5 has changed since the original submission, but the difference between each of those isoforms is transcription start/end sites. For the resubmission, we decided to focus more on splicing changes and less on differing TSS/TES, and so only included a single isoform for each unique intron chain (Methods). The equivalent figure would be S9.

5. Abstract: In the text “no difference between CLL SF3B1MT and B cell”, is there a mistake in the superscript, i.e., should it be “CLL SF3B1WT ? If not, please explain SF3B1MT.

This sentence has since been removed from the abstract, but is now reworked for clarity in the text on p7:

“Comparing the expression fold-change between CLL samples revealed that IR isoforms were globally down-regulated in the SF3B1^{K700E} sample compared to CLL SF3B1^{WT} (Fig. 4a). When performing the same comparison between B cell and SF3B1^{K700E}, we observed no significant difference in the expression of IR-containing isoforms (two-sided Mann-Whitney U p=0.121).”

The difference in global IR expression levels when comparing SF3B1MT and B cell is not significant. We see stronger downregulation of IR expression between SF3B1WT and SF3B1MT. The text has been changed to ‘no significant difference’ to provide more clarity than ‘no difference’.

Reviewer #3 (Remarks to the Author):

The manuscript by Tang et al describes FLAIR; a tool developed to analyze full length transcripts including detection and quantification of alternative isoforms from error prone long read sequencing data. To demonstrate the use of FLAIR, cDNA was generated in SF3B1 wild type and mutant chronic lymphocytic leukemia primary cells and long read sequencing data was acquired using Oxford Nanopore Technologies MinION sequencing. The paper presents what appears to be an extremely useful new tool and demonstrates its efficacy in a compelling biological context

1. There is less technical detail on the sequencing in the main paper than might be ideal. For example the ‘Nanopore Sequencing Statistics’ section, it is unclear if this data was generated from a single or multiple flowcells. The amount of sequencing generated is not detailed. And the gene coverage appears lower than might be anticipated. In addition, as the authors mentioned in the discussion, multiple runs may create a batch effect. Additionally, the greater than 2X difference in 2D pass reads between B cell and CLL should be addressed.

Table 1 now contains more information about the total number of reads generated from all the runs. The three pilot samples were sequenced on three separate flow cells. The difference in read numbers is due to variability in the quality of flow cells. Since there is variability between flow cells, running one sample per flow cell can create a flow cell batch effect despite the libraries being prepared together. Since the read depth is variable between runs, we have sequenced multiple biological replicates and we also perform upper quartile normalization on the read counts for each sample for our analyses. Batch correction was used for differential expression and splicing analysis with DRIMSeq.

2. The key improvements vs other tools/approaches are mentioned but not explored in much detail. This seems like a fundamental part of the paper so it was surprising to have to read the results section several times to make sure I didn’t miss more exposition. Figure 2C provides

more detail than the text and the tool looks good in these comparisons. However, I wasn't clear why coverage was displayed on the x axis since it seemed to contribute minimally to precision/recall. Finally, I was confused by the use of 'increased' when describing the sensitivity of the new tool ("higher" maybe?).

We agree that evaluation of FLAIR against other tools is extremely important, and we also recognize that there is a lack of comparable tools. We have updated the figure to compare FLAIR with the newest version of MandalorION and added Pinfish, Oxford Nanopore's own tool that is unpublished (no bioRxiv either). We have a short description of the MandalorION and Pinfish algorithms. We test the minimum coverage for a group of reads to be considered an isoform as it is a threshold that determines how confident in each nanopore read we are and how well each tool can potentially detect rare transcripts. We have changed the description of sensitivity from increased to higher. This figure panel changed from 2c to 2b. Additionally, we include a comparison of sensitivity and precision based on intron chains (Supplementary Figure 5).

3. Isoform usage (fig 4d and Supplemental 5): These figures represent FLAIR's ability to quantify isoforms in specific genes allowing for proportional analysis. However, the low number of isoform reads for each gene makes the reproducibility of these plots questionable. Replicating the experiment would increase this confidence.

We have increased the number of replicates and the depth of sequencing to remedy this.

4. Methods: PCR was used to add adapters to the cDNA. As PCR has the potential to introduce bias in quantification of the isoforms, the methods should explicitly state PCR parameters including amplification cycles.

We used the protocol developed by Byrne et al. 2017 without modifications. We have added the number of amplification cycles in the Methods for convenience (15 cycles).

5. Short-read validation: The authors state "removed any novel, nanopore-only splice sites from our analysis due to limited evidence of their validity." This seems to suggest the authors believe short reads are necessary to validate the findings of the long reads. However, it was stated in the manuscript that FLAIR can be used with or without the short read data. The utility of FLAIR without needing short reads would seem to be a significant advancement over current tools/methods. Can the authors clarify this and potentially add data to back up their position.

The vast majority of minimap2-aligned nanopore reads use canonical splice sites or are near canonical splice sites (Figure 2a). There are 685,002 (out of 5,787,229 total used by the nanopore reads) splice sites found exclusively in minimap2-aligned reads used more than 3 times. The minimap2-only splice sites are those that were >10 bp away from any splice site observed in short-read sequencing or annotations. We looked at the minimap2-only cases and think that many can be attributed to sequencing/alignment error and hesitate to include them without validation. For this reason, although short-read splice junctions are not required for FLAIR, if short-read splice junctions can't be provided, FLAIR would then require gene annotations. There have already been users of FLAIR that did not provide short read data and only gene annotation (e.g. Tiek et al.). The use of short-read splice junctions in FLAIR for our

study, however, is necessary given that we are looking at cancer samples, some with splicing factor mutation, that may result in aberrant splicing that are not present in annotation that we hope to capture.

6. Additional information: A table containing Number of genes sequenced, coverage, Gb of data, median length, number of genes with differential splicing etc. would be interesting and useful to help future users of FLAIR to decide to use a global or targeted RNAseq approach.

Table 1 has now been expanded to contain more useful stats, including your suggestions of # of genes covered, % transcript covered on average, gigabase, median length of reads per run.

7. The way the aberrant splicing landscape is described was confusing to me: in some cases, there seems to be the suggestion that the wild type cells have “aberrant” intron retention? Can you clarify this for the general reader.

In our original analysis, we observed elevated levels of intron retention in CLL SF3B1 WT compared to both Normal B cell and CLL SF3B1 K700E, suggesting CLL SF3B1 WT had an aberrant intron retention landscape. We no longer observe this trend to be statistically significant and have removed this point.

8. It is unclear how much the GO analysis adds to the manuscript as it is currently presented. The GO analysis has been updated with our new data. Its use in characterizing genes with unproductive, intron-retaining isoforms has been described in more detail in the discussion now.

Minor:

1. Abstract: ‘...and no difference between CLL SF3B1MT’. MT—we assume this should be WT?

This was answered in a previous question in this document (Reviewer 2 minor question #5) and the answer has been duplicated here:

This sentence has since been removed from the abstract, but is now reworked for clarity in the text on p7:

“Comparing the expression fold-change between CLL samples revealed that IR isoforms were globally down-regulated in the SF3B1^{K700E} sample compared to CLL SF3B1^{WT} (Fig. 4A). When performing the same comparison between B cell and SF3B1^{K700E}, we observed no significant difference in the expression of IR-containing isoforms (two-sided Mann-Whitney U $p=0.121$).”

The difference in global IR expression levels when comparing SF3B1MT and B cell is not significant. We see stronger downregulation of IR expression between SF3B1WT and SF3B1MT. The text has been changed to ‘no significant difference’ to provide more clarity than ‘no difference’.

2. Results: Flair Provides improved full-length isoform detection....: Figure 3C is referenced, however in this context 2C should have been referenced?

Changed to Figure 2.

3. Figure 4a: The P-value indicates significance, however displayed like this, the fold-change difference appears very small.

We have included the difference between the medians of both groups in the legend now.

4. Early clarification that a global RNAseq approach was taken should be considered for ease of understanding.

This has now been clarified on p3: “Of a large cohort of CLL patient tumor samples characterized using short-read RNA-Seq, we have resequenced six of these transcriptomes, globally, with nanopore technology: three with wild type *SF3B1* and three with the K700E mutation.”

5. Productive vs unproductive terminology. Can you provide further bibliographical support for this construct?

We have included the following citations using the productive/unproductive terminology:

(Lewis et al. 2003) “the splicing factor SC35 has been shown to autoregulate its expression through regulated unproductive splicing and translation (RUST) by generating NMD-targeted isoforms”

(Sun et al. 2010) “Alternative splicing can regulate gene expression by generating nonproductive isoforms, such as mRNAs that are retained in the nucleus or are subject to NMD” (Filichkin and Mockler 2012) “Evidence also suggests that many pre-mRNAs undergo unproductive alternative splicing resulting in incorporation of in-frame premature termination codons (PTCs)”

6. ‘Sites that align outside of the wiggle room’. Please define wiggle room

‘Wiggle room’ and ‘window size’ were used interchangeably and now the terminology is uniformly ‘window size’. Window size is clarified in the main text now, p4: “To correct splice sites, an incorrect splice site in a read alignment was replaced with a correct one as long as the correct splice site is within a window size of 10 bp.”

7. ‘which we resolve by smoothing gaps of 30 nt or fewer’ please clarify how this was accomplished.

We have changed the description of “smoothing gaps” now to explain that indels were removed from the alignments. For example, deletions were treated as matches to the reference. This is included in the Methods section now.

8. Statistical tests are quoted with p values but the comparator groups are hard to discern.

We have included p-values on the figures themselves to make the comparator groups easier to distinguish.

9. FLAIR workflow great and helps orient the reader. A similar panel demonstrating the experimental workflow may help the reader navigate the manuscript.

We have now added an additional panel to Figure 1 that shows the experimental workflow.

10. Generally the figure legends could use more detail allowing the reader to understand the figure alone without flipping back and forward to the text multiple times.

We have expanded figure legends more such that the reader can understand the figures with minimal context from the main text.

Reviewers' comments:

Reviewer #1 (Remarks to the Author):

In the revised version of the manuscript the authors added replicate samples and substantially increased sequencing read depth. They also included new analyses of existing short read data and provide additional clarifications. The manuscript is much improved and findings better supported. The authors have addressed my comments raised during the initial review.

Reviewer #2 (Remarks to the Author):

In this revised manuscript the authors have addressed my previous criticisms and have provided a substantial amount of additional raw data by sequencing more samples more deeply. Moreover, they have considerably more information comparing FLAIR to other methods, as requested by reviewer(s). I appreciate the apparent superiority of FLAIR with regard to sensitivity, and FLAIR's comparable or higher precision compared to other methods, but am not qualified to comment on these technical details. I do note that the use of an improved alternative splicing event caller together with the additional long read sequencing data allowed the authors to increase the number of alternative 3' and 5' splice sites detected in comparisons between SF3B1 K700E and wild type. Other good additions to the revised manuscript include the demonstration of FLAIR's ability to identify coordinated splicing events, e.g. in the ADTRP shown in Figure 4d-4e; and the finding of strong TGAC branch point motifs in introns that were significantly better spliced in SF3B1 K700E. (Does the bottom isoform in Supplementary Figure 9a also show coordination between intron retention and inclusion of a small exon not seen in other isoforms?).

The new data does raise some relatively minor issues that require clarification.

1. I'm a little confused about the use of FLAIR for alternative 3' splice site identification. Of the 65 alternative sites previously identified in a cohort of 37 CLL samples using short read data, at least 6 events were relatively highly represented in long read data (>25 reads) where they exhibited highly correlated dPSI values in comparison to the short read data (Figure 3A). However, subsequently the authors wrote and utilized an alternative splice site caller for FLAIR, to independently identify alternative 3' ss that differ between WT and K700E, and found 35 such sites. The puzzle is that only 2 of these overlap with the original 65 sites. Does this suggest that the new methodology has both strengths and weaknesses, to identify new splicing changes missed by other methods, but also miss some splicing changes found by the other techniques? Or is there another interpretation?

2. One comment on the regulation of intron retention events. Comparison of normal B cells, CLL SF3B1 WT and CLL SF3B1 K700E samples showed that the CLL WT vs K700E transcriptomes differed with respect to intron retention, and the authors concluded that the K700E mutation causes reduced intron retention. However, since retention in CLL SF3B1 WT vs normal B cells was similar, could an alternative interpretation be that CLL SF3B1 WT cells exhibit possess mutations(s) that increase intron retention?

3. Supplementary Figure 10 appears to require some editing in the figure legend. The legend says that unproductive isoforms are represented by hatched patterns, but the pink isoform (solid color) is said to be nonproductive due to NMD, while the green isoform (solid pattern) is said to be nonproductive due to nuclear detention, and the light blue isoform is described as productive even though it is hatched.

Minor issues:

1. Fig. 1B: Presumably, the red bars in the genome alignments indicate sequence errors. Please indicate this in the legend.

2. In a couple of places it appears that the numbers don't add up, if I am interpreting the data correctly.

p. 4 : 326,699 high confidence spliced isoforms were identified by FLAIR, a good fraction appear not to be represented by the following descriptions: 32,479 ~10% match annotated isoforms; 142,971 represent novel combinations of already annotated splice junctions; 21,700 differ by virtue of a retained intron; 3594 differ by having a novel exon). What do the other ~38% of the isoforms represent?

p. 7: A total of 77 retained introns were said to be differentially retained in SF3B1 K700E compared to SF3B1 WT CLL, but the data in Figure 4c only shows data for 69 such introns.

3. Figure 4c is slightly confusing because the text discusses the data in terms of the magnitude of down-regulation of intron retention, but the figure shows data for introns whose intron retention is up-regulated.

Reviewer #4 (Remarks to the Author):

This paper explores the utility of nanopore technology based long read sequencing for isoform detection and differential isoform expression analysis, applied on CLL samples. Samples with a mutation in the splicing factor SF3B1, associated with the generation of aberrant splicing patterns are compared to SF3B1 WT samples. A special pipeline has been put together and presented in this work: FLAIR, which allows the creation of consensus isoforms from the inherently high error nanopore reads. An approach to perform differential isoform expression calculation from long read data is also presented. Both reviewers of the original submission agree that this work is of high interest for the splicing community, -this reviewer would add that the work is of general interest for the genomics and transcriptomics community, as nanopore based long read sequencing is being increasingly used due to the wide spread of the technology and relatively inexpensive equipment costs compared to current alternatives.

In this revised version the authors worked towards addressing the comments of both reviewer #1 and #2, thus improving on the original submission. Figure 1 contains now clear design study and data processing information.

Most significant change is the generation of millions of additional reads by taking advantage of the PromethION instrument high throughput sequencing capability. Therefore, the splicing data are now much more convincing, as the entire set now includes nearly 150 million reads. This addition of data increased the statistical power of the study, and has been supplemented by a thorough overhaul and update of the analysis process, detailed in the rebuttal.

Overall the tool described in the manuscript would make a welcome addition to the current nanopore data analysis toolkits available to the genomics community.

Further specific points:

1. In the introduction (third paragraph page 2) the authors compare PacBio vs Nanopore technology. PacBio technology is able to produce higher accuracy CCS reads, however the authors point out that higher throughput, by an order of magnitude, can be achieved in the PromethION instrument. Here it is important to include some current figures that compare throughputs achievable/library sequenced per SMRT cell or flowcell respectively in Sequel I&II vs PromethION. One more comment would be useful: the relative power of one high quality read from one molecule vs multiple overlapping lower accuracy reads from independent molecules.

2. As described in the introduction and through the results, the authors used short read data to

validate the long read derived results. It would be very useful to include a comment in the discussion, as to whether Nanopore read and FLAIR combination is able to replace short read sequencing for isoform and differential isoform detection and under what conditions of sequencing depth. If short read data are needed in addition to long read data, again a read depth recommendation, based on the presented work would be useful. Such comments would make the utility of the technology and FLAIR clearer to a wider audience.

3. One point is missing in the methods/results section: What are the criteria used to distinguish between incomplete transcript processing and intron retention? This should be clarified.

4. In the methods section: What data processing took place after basecalling? Did the authors attempt to identify the smartseq TSO oligo sequences at the 5' and polyA sequences at the 3'? Reads without such sequences were included or excluded from the analysis? Please clarify.

We would like to thank the referees for the positive feedback regarding the additional sequencing and analysis that we performed. We thank the referees for another round of productive feedback. We have read and addressed the remaining concerns.

Reviewer #1 (Remarks to the Author):

In the revised version of the manuscript the authors added replicate samples and substantially increased sequencing read depth. They also included new analyses of existing short read data and provide additional clarifications. The manuscript is much improved and findings better supported. The authors have addressed my comments raised during the initial review.

Reviewer #2 (Remarks to the Author):

In this revised manuscript the authors have addressed my previous criticisms and have provided a substantial amount of additional raw data by sequencing more samples more deeply. Moreover, they have considerably more information comparing FLAIR to other methods, as requested by reviewer(s). I appreciate the apparent superiority of FLAIR with regard to sensitivity, and FLAIR's comparable or higher precision compared to other methods, but am not qualified to comment on these technical details. I do note that the use of an improved alternative splicing event caller together with the additional long read sequencing data allowed the authors to increase the number of alternative 3' and 5' splice sites detected in comparisons between SF3B1 K700E and wild type. Other good additions to the revised manuscript include the demonstration of FLAIR's ability to identify coordinated splicing events, e.g. in the ADTRP shown in Figure 4d-4e; and the finding of strong TGAC branch point motifs in introns that were significantly better spliced in SF3B1 K700E.

(Does the bottom isoform in Supplementary Figure 9a also show coordination between intron retention and inclusion of a small exon not seen in other isoforms?).

Yes, we only highlighted the intron retention that was called as significant but there is also a smaller, additional exon.

The new data does raise some relatively minor issues that require clarification.

1. I'm a little confused about the use of FLAIR for alternative 3' splice site identification. Of the 65 alternative sites previously identified in a cohort of 37 CLL samples using short read data, at least 6 events were relatively highly represented in long read data (>25 reads) where they exhibited highly correlated dPSI values in comparison to the short read data (Figure 3A). However, subsequently the authors wrote and utilized an alternative splice site caller for FLAIR, to independently identify alternative 3' ss that differ between WT and K700E, and found 35 such sites. The puzzle is that only 2 of these overlap with the original 65 sites. Does this suggest that the new methodology has both strengths and weaknesses, to identify new splicing changes

missed by other methods, but also miss some splicing changes found by the other techniques? Or this there another interpretation?

Yes, even though there were 6 events found from short-read sequencing that were highly expressed in the nanopore data, they were not all called as significant. Another factor, besides depth of coverage, that affects statistical significance is sample size. The Wang et al. Cancer Cell 2016 study and our study differ in that the short-read study had 37 CLL samples and our study contained 6, thus our study possesses less power to detect statistically significant alt 3' ss.

Fig 3A is mainly showing that for the short read-identified events that have coverage in the nanopore data, how concordant the PSIs are. Even though some short read-identified events were relatively highly expressed in long read and thus had the potential to be detected, those events were not included in the 35 nanopore alt3'ss set for reasons we found to be due to the filters we implemented for detecting alt 3' ss in the nanopore data; for example:

- events that also had significant p values in the long-read but the difference in PSI between wt and mt was < 10%
- the p values did not reach statistical significance in the long-read data. Another factor being the smaller sample size in our long-read study

We have added a sentence to the Discussion commenting on the small amount of overlap:

“While we were able to detect alternative 3’SSs recapitulating *SF3B1* biology, nanopore sequencing was not able to detect as many altered events as short-read sequencing potentially due to the smaller cohort and the difficulty of detecting subtle splicing alterations. The small overlap between nanopore-identified and short read-identified alternative 3’ss could also be due to the stringent filtering applied in an effort to determine alternative splicing more accurately.”

2. One comment on the regulation of intron retention events. Comparison of normal B cells, CLL SF3B1 WT and CLL SF3B1 K700E samples showed that the CLL WT vs K700E transcriptomes differed with respect to intron retention, and the authors concluded that the K700E mutation causes reduced intron retention. However, since retention in CLL SF3B1 WT vs normal B cells was similar, could an alternative interpretation be that CLL SF3B1 WT cells exhibit possess mutations(s) that increase intron retention?

The referee brings up a good point, that it could be possible to frame this as an increase in IR in the CLL SF3B1 WT or a decrease in IR in CLL SF3B1 K700E. Previous literature has shown that CLL has elevated levels of splicing alterations, regardless of SF3B1 mutation status (Hacken et al. 2018, Yin et al. 2019). For our study, we wanted to focus on the subset of introns that are more spliced out in K700E. The K700E-preferred branch point sequence we found to be associated with these intron retentions supports that there is a subset of introns that will exhibit decreased intron retention in the mutant. To conclude, both interpretations are in part true, i.e.

CLL has increased intron retention and CLL SF3B1 MT contains introns that experience less retention.

We have added this to the Discussion:

“In addition, long-read sequencing highlighted a downregulation of isoforms containing retained introns in *SF3B1*^{K700E} relative to *SF3B1*^{WT}. This downregulation was corroborated by reanalyzing CLL, Nalm-6 cell lines, and TCGA BRCA short-read datasets with mutant *SF3B1*. CLL has been shown to contain elevated levels of splicing alterations, regardless of SF3B1 mutation status^{16,65}. The subset of introns that exhibit increased splicing in the mutant point to a different intron retention landscape in CLL *SF3B1*^{K700E}. Introns more significantly spliced out in *SF3B1* mutated samples contained a strong branch point TGAC sequence ~15 bp upstream of the 3' splice site, consistent with previously reported branch site motifs of altered 3' splice sites associated with the mutation²².”

3. Supplementary Figure 10 appears to require some editing in the figure legend. The legend says that unproductive isoforms are represented by hatched patterns, but the pink isoform (solid color) is said to be nonproductive due to NMD, while the green isoform (solid pattern) is said to be nonproductive due to nuclear detention, and the light blue isoform is described as productive even though it is hatched.

The fill patterns for the exons correspond to productivity as predicted by the NMD rule. In the legend, we compare the predicted productivity with the experimentally validated productivity for the same isoforms. To prevent confusion, we have clarified this in the legend now and refer to the isoforms by their name. The legend for SFig 10 now reads:

“The NMD-rule predicted productivity for the isoforms ENST00000581979.5 (purple), 37d6af42 (yellow, referred to as Isoform V in Sun et al. 2010), and ENST00000258962.4 (light blue) match the experimentally determined productivity determined in Sun et al. 2010. ENST00000581979.5 is detained in the nucleus and the Isoform V is unproductive via degradation by NMD. The remaining isoforms were not included in previous productivity studies.”

And for convenience, the sentence in the main text that refers to this figure (P8) is here:

“For example, *SRSF1* has several unproductive transcripts that are known to be either nuclear-retained or NMD-triggering⁶⁰, two of which (ENST00000581979.5 and unannotated Isoform V⁶⁰) we were able to identify and accurately predict as unproductive in our nanopore data (**Supplementary Fig. 10**). “

Minor issues:

1. Fig. 1B: Presumably, the red bars in the genome alignments indicate sequence errors. Please indicate this in the legend.

This has been clarified. The legend for Fig 1B now reads:

“First, reads are aligned to the genome with a spliced aligner. The sequence errors are marked in red. Next, they are splice-corrected using splice sites from either annotated introns, introns from short-read data, or both.”

2. In a couple of places it appears that the numbers don't add up, if I am interpreting the data correctly.

p. 4 : 326,699 high confidence spliced isoforms were identified by FLAIR, a good fraction appear not to be represented by the following descriptions: 32,479 ~10% match annotated isoforms; 142,971 represent novel combinations of already annotated splice junctions; 21,700 differ by virtue of a retained intron; 3594 differ by having a novel exon). What do the other ~38% of the isoforms represent?

The remaining isoforms that were not categorized are those that contain unannotated splice site usage that was supported by short reads. We have included this on P4 now:

“Most of the unannotated isoforms were a novel combination of already annotated splice junctions (142,971), while others deviated from the annotation because they contained a retained intron (21,700) or a novel exon (3,594). The remainder of the unannotated isoforms contain at least one novel splice site not present in annotations but supported through short reads.”

p. 7: A total of 77 retained introns were said to be differentially retained in SF3B1 K700E compared to SF3B1 WT CLL, but the data in Figure 4c only shows data for 69 such introns.

Upon revisiting the analysis that was performed, we realize that 77 is a typo; 77 was a number from an older version of the analysis and that number was not updated when the analysis was updated.

The number of significant IR identified is 70 for the newest version of analysis that was used in the manuscript. We can confirm from the analysis files and also from manually counting all the points that there are 70 introns plotted in 4C and SFig 8A. We have corrected the miscounted “n=” in Figure 4C as well as the main text to reflect the analysis containing 70 introns.

3. Figure 4c is slightly confusing because the text discusses the data in terms of the magnitude of down-regulation of intron retention, but the figure shows data for introns whose intron retention is up-regulated.

Yes, Figure 4c shows introns that exhibit increased retention in the mutant ($dPSI > 10$) as well as the introns that exhibit decreased intron retention in the mutant ($dPSI < -10$). The magnitude of change (the absolute value of the $dPSI$) is significantly greater for the $dPSI < -10$ group than the $dPSI > 10$ group. We have added more clarity to the text on P7:

“Although there were fewer significant introns found to be downregulated in the mutant ($\Delta\text{PSI} < -10$), the magnitude of downregulation was stronger for those introns (**Fig. 4c, Supplementary Table 1**).”

Reviewer #4 (Remarks to the Author):

This paper explores the utility of nanopore technology based long read sequencing for isoform detection and differential isoform expression analysis, applied on CLL samples. Samples with a mutation in the splicing factor SF3B1, associated with the generation of aberrant splicing patterns are compared to SF3B1 WT samples. A special pipeline has been put together and presented in this work: FLAIR, which allows the creation of consensus isoforms from the inherently high error nanopore reads. An approach to perform differential isoform expression calculation from long read data is also presented. Both reviewers of the original submission agree that this work is of high interest for the splicing community, -this reviewer would add that the work is of general interest for the genomics and transcriptomics community, as nanopore based long read sequencing is being increasingly used due to the wide spread of the technology and relatively inexpensive equipment costs compared to current alternatives.

In this revised version the authors worked towards addressing the comments of both reviewer #1 and #2, thus improving on the original submission. Figure 1 contains now clear design study and data processing information.

Most significant change is the generation of millions of additional reads by taking advantage of the PromethION instrument high throughput sequencing capability. Therefore, the splicing data are now much more convincing, as the entire set now includes nearly 150 million reads. This addition of data increased the statistical power of the study, and has been supplemented by a thorough overhaul and update of the analysis process, detailed in the rebuttal.

Overall the tool described in the manuscript would make a welcome addition to the current nanopore data analysis toolkits available to the genomics community.

Further specific points:

1. In the introduction (third paragraph page 2) the authors compare PacBio vs Nanopore technology. PacBio technology is able to produce higher accuracy CCS reads, however the authors point out that higher throughput, by an order of magnitude, can be achieved in the PromethION instrument. Here it is important to include some current figures that compare throughputs achievable/library sequenced per SMRT cell or flowcell respectively in Sequel I&II vs PromethION. One more comment would be useful: the relative power of one high quality read from one molecule vs multiple overlapping lower accuracy reads from independent molecules.

The technologies offered by PacBio and ONT are rapidly evolving, which we touched upon in the discussion. At the time of sequencing our samples (January-February 2019), we were not yet aware of the Sequel II (PacBio blogpost announcing the Sequel II is dated April 24, 2019 <https://www.pacb.com/blog/now-available-sequel-ii-system-delivers-8-times-as-much-data-as-previous-system/>). There appear to be few or no papers that have published on Sequel II data, particularly transcriptome sequencing.

Whichever technology can reliably produce high quality, high throughput reads for cheap and uses less starting material would of course be the most powerful for isoform discovery and quantification. However, for the purposes of calculating isoform abundances and comparing isoform expression between conditions, multiple lower accuracy reads would be more informative than fewer high quality reads. With the Sequel 2's increase in throughput, having to choose between quantity and quality may no longer be the case. While this is an important tipping point, we have not used the Sequel 2 yet and do not want to comment too much further on PacBio technology. In any case, we have added to the discussion two new citations of PacBio's papers on bioRxiv demonstrating the throughput of their sequel II for genome sequencing.

Nanopore still has the advantage of giving users the option of PCR-free preps (native RNA sequencing), sequencing base modifications (native RNA or DNA sequencing), higher accuracy cDNA sequencing (R2C2 sequencing, Volden et al. 2018), and, more important for genome assemblies, longer read lengths in general (longest 2.2 megabases, Payne et al. 2018).

As such, we commented on all of this in the Discussion (P10):

"Fortunately, the throughput and accuracy of nanopore and PacBio technology has the potential to increase with subsequent iterations of the technologies⁶⁹. For nanopore in particular, methods to achieve higher sequence accuracy³⁶ or circumvent PCR bias and reverse transcription length restrictions³⁰ have been developed. In line with the rigorous pace of improvements in the field of long reads, PacBio has recently improved their throughput 8X with the newest PacBio Sequel II system, which has been shown to generate ~19 and 83 Gb of consensus reads^{38,39}."

And the original, erroneous sentence in the introduction stating that the PromethION could produce orders more magnitude of reads than pacbio has been removed and replaced with (P11):

“The high-throughput Oxford Nanopore PromethION has the potential to yield upwards of 100 gigabases (Gb) per flow cell⁴⁰; the substantial number of long molecules sequenced makes the PromethION applicable to our purposes of transcript detection and quantification.”

2. As described in the introduction and through the results, the authors used short read data to validate the long read derived results. It would be very useful to include a comment in the discussion, as to whether Nanopore read and FLAIR combination is able to replace short read sequencing for isoform and differential isoform detection and under what conditions of sequencing depth. If short read data are needed in addition to long read data, again a read depth recommendation, based on the presented work would be useful. Such comments would make the utility of the technology and FLAIR clearer to a wider audience.

We state in the discussion that obtaining as much sequencing as possible is ideal since the saturation analysis reveals that there are still more isoforms that can potentially be sequenced. We also state in the discussion that nanopore should be used with short read for more confident isoform calling. We also include a more quantitative point regarding read depth, that there should be at least 3 replicates. P10:

“A subsampling analysis revealed that we have not saturated the number of discoverable isoforms. Despite efforts to obtain nanopore sequencing data from a more high-throughput sequencing platform (PromethION) and account for the low accuracy of 1D nanopore sequencing, we note that the read depth [is still one of the] limiting factors of this study. [...] Everything considered, studying splicing factor mutations in primary patient samples using nanopore sequencing with fewer reads than the current study or without short-read sequencing would be suboptimal. Short-read sequencing was necessary for increasing confidence in splice sites, although future work with higher accuracy reads could potentially obviate the need for short reads. Future studies of primary samples should also include larger cohort sizes, with three samples per genotype being the minimum. Even though short-read technology is able to sequence more deeply than long reads, the ability of short reads to saturate splice junction detection is depth-dependent⁶⁹; thus, splicing studies should aim to sequence as deeply as possible.”

As always, researchers should try to obtain as much sequencing as possible (for long and short read). Thus, we did not specify a specific depth we would recommend. Power analyses and saturation curves using short reads have historically shown that more sequencing would bolster more discoveries. For example, the intropolis study of all human reads in SRA show that with

more data, more splice junctions can be discovered (Nellore et al 2016). The figure is copied here for convenience.

The authors measured the accumulation of novel human splice junctions corresponding to new submissions to the SRA. Although they note that the confidently called splice junctions are almost saturated, the amount of sequencing necessary to reach saturation is still a large amount; in addition, they note that by sequencing new samples, particularly total RNA samples, diverse splice junctions can still be discovered.

3. One point is missing in the methods/results section: What are the criteria used to distinguish between incomplete transcript processing and intron retention? This should be clarified.

These are poly(a)-selected so we assumed that the transcripts for the most part are fully processed. No criteria were used to distinguish between incomplete transcript processing and intron retention in the interest of simplifying the analysis pipeline. We feel that attempting to distinguish incompletely processed transcripts from deliberately retained introns is outside of the scope of this work and will be of interest for future improvements to FLAIR. We would also note that we are interested in identifying splicing alterations due to CLL and/or the SF3B1 mutation; therefore, we would prefer to sequence incomplete transcripts that might have not been fully processed due to aberrant splicing as well as completely processed transcripts that ended up with intron retention. We now state in the methods explicitly that we do not perform further categorization of intron retentions (P13):

“For differential isoform usage testing (FLAIR *diffExp*), isoforms were grouped by gene and only genes with at least 25 reads in 4 of the 6 samples were tested. As these libraries were poly(A) selected, we did not distinguish between intron retentions due to

incomplete transcript processing and intron retentions deliberately retained due to sample genotype.”

4. In the methods section: What data processing took place after basecalling? Did the authors attempt to identify the smartseq TSO oligo sequences at the 5' and polyA sequences at the 3'? Reads without such sequences were included or excluded from the analysis? Please clarify.

We included all pass reads in the analysis, including those which did not contain smartseq adapters on both ends. FLAIR is equipped to deal with potentially truncated reads. Information can be gleaned from truncated reads with sufficient length to be assigned to an isoform; reads that are too short for a unique assignment are not included in the isoform quantification. We clarify this in the methods now in the isoform identification section (P12):

“All pass reads, including reads that did not contain sequenced adapters on both ends, were used when running FLAIR...”

And in more detail in the data handling section (P11):

“We identified reads with adapter sequences on both ends following the approach employed in the MandalorION pipeline³⁴: (1) adapters are aligned to all the reads using blat⁷³, (2) if there are at least 10 bases at the left and right ends of the reads that match the adapter sequence then the read is considered to have adapters on both ends. We found that only a fraction of our reads that were called as “pass” reads contained the adapter sequences on both ends (~35-55%). In the interest of being able to use more of our data, we did not remove these reads from the analyses.”